# Private Statistical Estimation via Truncation

**Manolis Zampetakis**
Yale University
manolis.zampetakis@yale.edu

**Felix Zhou**
Yale University
felix.zhou@yale.edu

## Abstract

We introduce a novel framework for differentially private (DP) statistical estimation via data truncation, addressing a key challenge in DP estimation when the data support is unbounded. Traditional approaches rely on problem-specific sensitivity analysis, limiting their applicability. By leveraging techniques from truncated statistics, we develop computationally efficient DP estimators for exponential family distributions, including Gaussian mean and covariance estimation, achieving near-optimal sample complexity. Previous works on exponential families only consider bounded or one-dimensional families. Our approach mitigates sensitivity through truncation while carefully correcting for the introduced bias using maximum likelihood estimation and DP stochastic gradient descent. Along the way, we establish improved uniform convergence guarantees for the log-likelihood function of exponential families, which may be of independent interest. Our results provide a general blueprint for DP algorithm design via truncated statistics.

## 1 Introduction

In an era of data-driven decision-making, differential privacy (DP) has become the gold standard for privacy-preserving statistical analysis, ensuring that the inclusion or exclusion of any individual's data does not significantly alter outcomes [20]. DP has seen widespread adoption, including in the U.S. Census Bureau's data releases [2] and industry applications [21], due to its rigorous guarantees balancing privacy and utility.

Over the past two decades, research has produced private estimation methods for mean estimation [45], regression [44], and hypothesis testing [24]. Techniques like the Laplace and Gaussian mechanisms [20], differentially private empirical risk minimization [13], and DP-SGD [1] have enabled privacy-preserving machine learning.

Despite this long line of work, a key limitation remains: there is no general-purpose computationally efficient framework for differentially private statistical estimation, when the support of the data is unbounded, e.g., $\mathbb{R}^d$. In such cases, bounding the *sensitivity* of the statistical estimators is a very challenging task and existing methods require case-specific sensitivity analyses, making their broad application challenging. Without a good bound on the sensitivity of an estimator, it is impossible to obtain good DP mechanisms with utility-privacy tradeoffs.

**Data Truncation.** One natural approach to reducing sensitivity—and thereby improving privacy-utility trade-offs—is to *artificially truncate the data*, ensuring that extreme values do not unduly influence the estimation process. Truncation directly controls sensitivity, which is crucial in DP settings where privacy guarantees depend on bounding the worst-case impact of a single data point. While this technique provides a compelling solution to the challenge of sensitivity control, it introduces a new issue: *bias*. Truncation distorts the underlying distribution, leading to inaccurate estimates if not properly corrected. This raises an important question:

*Can we leverage techniques from truncated statistics to develop a principled framework*

39th Conference on Neural Information Processing Systems (NeurIPS 2025).

The study of truncated statistics has a long history, with recent results providing efficient methods for estimating distributions and regression models under truncation. Notably, works such as Daskalakis et al. [17] have developed polynomial-time algorithms for estimating Gaussian parameters from truncated samples, overcoming computational barriers. However, despite the rich theory of truncated statistics, its potential for designing differentially private estimators remains largely unexplored. In this work, we take a step in this direction by introducing a principled approach that integrates differential privacy with statistical methods designed for truncated data. A novel consequence of our work is the **first efficient DP algorithm for estimating the parameters of unbounded high-dimensional exponential families**.

## 1.1 Contributions

Our main conceptual contribution is a method for private statistical estimation using truncation. Using this paradigm, we design the first algorithm for privately estimating the parameter of unbounded high-dimensional exponential family distributions. As special cases, we recover algorithms for Gaussian mean and covariance estimation with near-optimal sample complexities.

We write $m$ to denote the dimension of the parameters/sufficient statistics of an exponential family distribution $q_\theta$ parameterized by $\theta$, and $d$ to denote the dimension of the distribution. Let $\varepsilon, \delta \in (0, 1)$ denote the approximate-DP parameters and $\alpha \in (0, 1)$ denote the accuracy parameter. We design the following efficient $(\varepsilon, \delta)$-DP algorithms that outputs:

- (See **Theorem 3.1**) an estimate $\hat{\theta}$ for the parameter of an exponential family distribution $q_{\theta^\star}$ such that $\|\hat{\theta} - \theta^\star\| \leq \alpha$ with sample complexity that is linear in $m$ and proportional to $1/\varepsilon$, $1/\alpha^2$, and $1/\alpha\varepsilon$.

- (See **Theorem 4.1**) an estimate $\hat{\mu}$ for the mean of a Gaussian $\mathcal{N}(\mu^\star, I)$ such that $\|\hat{\mu} - \mu^\star\| \leq \alpha$ with sample complexity that is linear in $d$ and proportional to $1/\varepsilon$, $1/\alpha^2$, and $1/\alpha\varepsilon$.

- (See **Theorem 4.2**) an estimate $\widehat{\Sigma}$ for the covariance of a Gaussian $\mathcal{N}(0, \Sigma^\star)$ such that $\|I - (\Sigma^\star)^{-\frac{1}{2}}\widehat{\Sigma}(\Sigma^\star)^{-\frac{1}{2}}\|_F \leq \alpha$ with sample complexity that scales as $d^2$ and is proportional to $1/\varepsilon$, $1/\alpha^2$, and $1/\alpha\varepsilon$.

In particular, Theorem 3.1 is the first efficient algorithm for privately estimating unbounded high-dimensional exponential families. All prior works consider only bounded or one-dimensional/one-parameter cases (see related works in Section 1.2). Theorems 4.1 and 4.2 demonstrate the sample efficiency of our method by recovering the optimal sample complexities for the specific case of Gaussian estimation.

**Technical Contributions.** The key idea in obtaining our private estimator for exponential families is to only access data after truncating to an appropriate bounded survival set. Then, folklore techniques suffice to estimate the parameters of the *truncated* distribution. This raises two issues: 1) there may be bias introduced by the truncation and 2) how can we choose the survival set?

In Sections 3.2 to 3.5, we first assume we are given a bound $R = O(1)$ on the radius of the norm of the true parameter so that one straightforward choice for the survival set is the ball of radius $O(\sqrt{m})$ about the origin. Then we address 1) by using stochastic gradient descent (SGD) on the *truncated* negative log-likelihood function $L$ over a carefully chosen projection set $K$ to ensure strong convexity. The true parameter is a minimizer of $L$. However, to satisfy privacy, we must use DP-SGD, which requires making multiple passes over the data. This raises further issues since each truncated sample only provides a single unbiased gradient estimate. We overcome this by instead optimizing the *empirical* log-likelihood $\tilde{L}$, which necessitates a uniform convergence result to ensure that the empirical minimizer remains close to the population minimizer.

Shah, Shah, and Wornell [43] also derived a uniform convergence result for exponential families but their proof does not immediately handle truncation. Furthermore, they require $O(1/\alpha^4)$ samples, which would lead to sub-optimal sample complexity for Gaussian mean and covariance estimation. Our proof overcomes this limitation by first showing that $\tilde{\theta}^\star \in K$ lies in the projection set of $\tilde{L}$ after $O(1)$ samples. Hence $\tilde{L}$ actually satisfies a *Polyak-Lojasiewicz (PL)* condition, which leads to uniform convergence at $O(1/\alpha^2)$ samples.

To address 2), we observe that simply estimating the parameter of the truncated distribution yields a constant-distance warm start. In Appendices B.6 and B.7, we remove the need for a prior by adapting a standard bounding box algorithm for the parameter of the truncated distribution that attains a $O(\text{poly}(m))$-distance warm start. This is adapted from a folklore algorithm for Gaussian estimation that we generalize to truncated exponential families. Next, to avoid unnecessary $\text{poly}(m)$-dependence in the sample complexity, we further refine this to an $O(1)$-distance warm start by adapting a Gaussian estimation algorithm of Biswas et al. [7] that we generalize to truncated exponential families.

Finally, in Section 4, we show that we can derive algorithms for Gaussian mean and covariance estimation from our general algorithm for exponential families. However, to avoid linear dependence on the condition number, we adapt a recursive Gaussian preconditioning algorithm by Biswas et al. [7] which we again generalize to the setting of truncated Gaussians.

## 1.2 Related Works

**DP Exponential Family Estimation.**  Prior works on privately estimating the parameter of exponential families focus either on asymptotic guarantees [23], bounded exponential families [6], or one-dimensional/one-parameter exponential families [6, 37]. In contrast, our algorithms can handle unbounded high-dimensional multi-parameter exponential families.

**DP Gaussian Estimation.**  The first sample-optimal DP Gaussian mean/covariance estimation algorithms were due to Aden-Ali, Ashtiani, and Kamath [3], who attains rates of $\tilde{O}(\frac{d}{\alpha^2} + \frac{d}{\alpha\varepsilon} + \frac{\log(1/\delta)}{\varepsilon})$ and $\tilde{O}(\frac{d^2}{\alpha^2} + \frac{d^2}{\alpha\varepsilon} + \frac{\log(1/\delta)}{\varepsilon})$, respectively. However, their algorithms require exponential running time. Recent transformations from robust algorithms to private algorithms obtained the same optimal sample complexities for mean estimation [26] and covariance estimation [27] in polynomial time. See Hopkins et al. [27, Table 1, Table 2] for a detailed summary of prior algorithmic results. The sample complexities are tight up to logarithmic factors [29, 30, 39, 42].

The private Gaussian estimation algorithms of Biswas et al. [7] and Karwa and Vadhan [30] are also relevant as we adapt them for truncated exponential families. Note that due to the bias introduced by the truncation step, directly running the adaptations on truncated samples yields biased estimates. However, we show that these biased estimates suffice as warm starts/preconditioners.

**DP Empirical Risk Minimization.**  A related line of work develops methods to solve empirical risk minimization problems (see e.g. Bassily, Smith, and Thakurta [5] and references therein). This line of work only handles the sensitivity problem that we describe above when the support of the input distribution is bounded. In the problem of learning exponential families that we explore in this paper, these assumptions are often violated and this is one illustration of the importance of the methods that we propose.

**Truncated Statistics.**  The recent seminal work of Daskalakis et al. [17] developed the first polynomial-time algorithms for estimating Gaussian parameters from truncated samples within a given survival set. This has led to a flurry of developments, including generalizations to truncated exponential families [34, 35], truncated Gaussian estimation with *unknown* survival sets [32, 35], truncated regression [15, 16, 18], and truncated linear dynamics [41].

## 2 Preliminaries

We include the standard preliminaries for differential privacy in Appendix A.

### 2.1 Notation

We write $d$ for the dimension of the ambient space, $m$ for the dimension of the sufficient statistic for an exponential family distribution $q_\theta$ parameterized by $\theta$, $\varepsilon, \delta$ for the privacy parameters, and $\alpha, \beta$ for the accuracy, failure probability parameters. We typically use $\rho \in (0,1)$ to indicate the survival probability when truncating a distribution to a survival set $S \subseteq \mathbb{R}^d$. For $R > 0$, we use $B_R(x)$ to denote the closed Euclidean ball of radius $R$ about $x$, or $B_R(\mathcal{X})$ to denote the union of closed Euclidean balls of radius $R$ about $x \in \mathcal{X}$. $B_{-R}(\mathcal{X}) := \{x \in \mathcal{X} : B_R(x) \subseteq \mathcal{X}\}$ denotes $R$-interior of

the argument. We let $\mathbb{S}^d$ denote the set of $d \times d$ real-symmetric matrices. For $A, B \in \mathbb{S}^d$, we write $A \prec B, A \preceq B$ to denote the positive definite and positive semi-definite relations. $\mathbb{S}^d_+$ indicates the subset of positive semi-definite matrices.

## 2.2 Neighboring Truncated Datasets

Our guiding principle for designing private algorithms is to discard outlier samples that fall outside some survival set[1] $S$ to obtain bounded sensitivity. However, given an algorithm $\mathcal{A}$ that is differentially private on truncated samples, it is not clear how to reason about the privacy guarantees when we first truncate a dataset and feed it to $\mathcal{A}$, since the truncated datasets may no longer be neighboring. Specifically, let $D'$ be obtained from the dataset $D$ by modifying an entry. Then, depending on whether the modified entry falls in $S$, the truncated datasets $D_S, D'_S$ fall into one of three categories: Either 1) $D'_S = D_S$, 2) $D'_S$ can be obtained from $D_S$ by modifying an entry, or 3) $D'_S$ can be obtained from $D_S$ by adding/deleting an element.

Below, we formally state the guarantees of a preprocessing procedure we employ in addition to truncation and defer its proof to Appendix A.3.

**Lemma 2.1.** *Fix $n \geq 1$ and let $N \in \mathbb{N}$. Let $D \in \mathbb{R}^{d \times N}$ be an $N$-sample dataset, $D' \in \mathbb{R}^{d \times N}$ be obtained from $D$ by modifying a single entry, and $S \subseteq \mathbb{R}^d$. Write $D_S, D'_S$ to denote the datasets obtained from $D, D'$ by discarding entries that fall outside of $S$. There is a preprocessing algorithm $\mathcal{A}$ such that*

   *(i) $\mathcal{A}(D'_S) \in S^n$ can be obtained from $\mathcal{A}(D_S) \in S^n$ by modifying a single element,*

   *(ii) for any $(\varepsilon, \delta)$-DP algorithm $\mathcal{B}$ with respect to neighboring truncated datasets, the composition $\mathcal{B}(\mathcal{A}(D_S))$ is $(\varepsilon, \delta)$-DP with respect to neighboring (untruncated) datasets $D, D'$, and*

   *(iii) if $D$ is sampled i.i.d. from some distribution $p$ such that $p(S) =: \rho$, then $\mathcal{A}(D)$ contains i.i.d. samples from the truncated distribution $p^S$ with probability $1 - \beta$, provided $N = \Omega(\frac{n \log(1/\beta)}{\rho})$.*

Throughout this work, we only work with truncated distributions with constant survival mass $\rho \geq \Omega(1)$. Thus in light of Lemma 2.1, it suffices to analyze the privacy and sample complexity of datasets from the truncated distribution and then incur a sample complexity blowup of $O(\log(1/\beta))$ for the untruncated distribution while maintaining the privacy guarantees. From hereonforth, we do not distinguish between the sample complexity of truncated and untruncated samples, but it is understood that we perform this preprocessing to obtain truncated samples. See e.g. Algorithm B.1 for an explicit example.

## 2.3 Exponential Families

Exponential families form a fundamental class of probability distributions that unify and generalize many statistical models, including the Gaussian, Bernoulli, and Poisson distributions [9]. Their structured mathematical form provides a natural framework for efficient statistical inference, enabling widespread applications in machine learning, information theory, and Bayesian statistics [49].

In this paper, we consider absolutely continuous[2] exponential family distributions over $\mathbb{R}^d$

$$q_\theta(x) = h(x) \exp\left(\theta^\top T(x) - \Upsilon(\theta)\right),$$

where $h : \mathbb{R}^d \to \mathbb{R}_+$ is the base measure, $T : \mathbb{R}^d \to \mathbb{R}^m$ is a sufficient statistic, $\Upsilon(\theta) = \log \int_x h(x) \exp(\theta^\top T(x)) dx$ is the *log-partition function* that ensures $q_\theta$ integrates to 1, and $\theta \in \mathcal{H} := \{\theta : A(\theta) < \infty\}$ is the *natural parameter* of the distribution.

It is not hard to see that $\mathcal{H}$ is convex. We write $\Theta \subseteq \mathcal{H}$ to be a closed, convex subset of the natural parameter space. As we illustrate in Section 4 for Gaussians, it may be necessary to restrict ourselves to $\Theta$ in order to obtain useful properties such as strong convexity of the NLL function.

---

[1]Daskalakis et al. [17] also referred to this as the *truncation set*.

[2]Our algorithms are able to handle more general distributions for which the empirical log-likelihood function is almost surely differentiable, but we present the case of absolutely continuous distributions with respect to the Lebesgue measure for simplicity.

**Statistical Assumptions.** We now state some common statistical assumptions for estimating (truncated) exponential families. Remark that these are necessary assumptions that appear even in the non-private literature for exponential families [11, 34, 35, 43]. Moreover, these assumptions hold for a variety of common distributions such as exponential distributions, Weibull distributions, continuous Bernoulli distributions, and continuous Poisson distributions [34, Appendix B].

**Assumption 2.2** (Statistical Assumptions).

(S1) (Bounded Condition Number) $\lambda I \preceq \mathrm{Cov}_{x \sim q_\theta}[T(x), T(x)] \preceq I$ for every $\theta \in \Theta$. (e.g. isotropic Gaussian satisfies this with $\lambda = 1$)

(S2) (Interiority) $\theta^\star$ is in the $\eta$-*interior* $B_{-\eta}(\Theta)$ of $\Theta$ for some $\eta \in (0, 1]$, i.e. $B_\eta(\theta^\star) \subseteq \Theta$.

(S3) (Log-Concavity) Each $q_\theta, \theta \in \Theta$ is a log-concave distribution (e.g. isotropic Gaussian)

(S4) (Polynomial Sufficient Statistics) $T(x)$ is a polynomial of given constant degree $k = O(1)$ (e.g. $k = 1$ for isotropic Gaussian).

Some remarks about the statistical assumptions are in order.

We can relax Assumption (S1) to $\lambda I \preceq \mathrm{Cov}[T(x), T(x)] \preceq \Lambda I$ by rescaling the sufficient statistics $T'(x) \leftarrow {}^{T(x)}/\sqrt{\Lambda}$ if necessary. Then the condition number becomes $\Lambda/\lambda$. In order to obtain computationally efficient algorithms, some assumptions on the spectrum of $\mathrm{Cov}[T(x), T(x)]$ are made even for non-privately learning exponential families [43]. Typically, as in the case of Gaussians (Appendix C.4), it is possible to precondition the distribution so that $\lambda = \Omega(1)$. Hence we think of $\lambda$ as being a constant bounded away from 0.

Assumption (S2) is usually easy to satisfy just by "blowing up" $\Theta$ by $\eta$ (e.g. Appendices C.1 and C.3 for Gaussians). Thus throughout this work, we think of $\eta$ as a small constant bounded away from 0. Assumptions (S1) and (S2) together imply a subexponential concentration inequality on the sufficient statistic $T(x)$ for $x \sim q_{\theta^\star}$ (Proposition A.8).

Assumptions (S3) and (S4) together imply an anti-concentration result of polynomials under log-concave measures [12]. This is a crucial ingredient in deriving computationally efficient algorithms for truncated statistics which appears even in the most basic case of learning truncated Gaussians [17]. Alternatively, we may assume that the sufficient statistics belong to a class of functions that satisfy anti-concentration. For simplicity of exposition, we focus on the case where $T(x)$ is a polynomial.

**Computational Subroutines.** In order to efficiently implement our algorithms, we will need access to a few problem-specific subroutines. We emphasize that these are standard computational tasks and specify how they can be achieved in the case of Gaussian mean and covariance estimation in Appendices C.1 and C.3.

**Assumption 2.3** (Computational Subroutines).

(C1) (Projection Acess to Convex Parameter Space) There is a $\mathrm{poly}(m)$-time projection oracle to $\Theta \ni \theta^\star$.

(C2) (Sample Access to Log-Concave Distribution) For every $\theta \in \Theta$, we can (approximately) sample from $q_\theta$ in $\mathrm{poly}(d)$-time.

(C3) (Moment-Matching Oracle) There is an oracle `MomentMatch` such that given some $\tau \in \mathbb{R}^m$, the oracle returns some $\theta \in \Theta$ such that $\mathbb{E}_{x \sim q_\theta}[T(x)] = \tau$ (approximately) holds in $\mathrm{poly}(m)$-time.

We also comment on the problem-specific computational subroutines required by our algorithm.

For simple convex sets like $\mathbb{R}^m$, half-spaces, Euclidean balls, or hypercubes, there are simple subroutines to compute the convex projection onto them. For general convex bodies, it suffices to assume access to a separation oracle in order to call on the ellipsoid method [25] and implement a projection oracle. Thus Assumption (C1) is typically not a strong condition. Moreover, our algorithm may require taking projections onto the intersections of closed convex sets which occur when we iteratively reduce the domain of the feasible region. This can be efficiently implemented via Djikstra's algorithm [8, 50].

There are efficient algorithms to sample from log-concave distributions under additional regularity conditions on $q_\theta$ [14] or when the support is a convex body [36]. Thus Assumption (C2) is not a stringent concern.

Finally, there are closed-form solutions for Assumption (C3) in many scenarios such as the standard parameterization of Gaussian distributions as an exponential family. In general, a moment matching oracle can be implemented by solving a maximum-likelihood estimation (MLE) problem, which is a convex optimization problem with efficient solutions (see e.g. Lee, Wibisono, and Zampetakis [34]).

## 2.4 The Negative Log-Likelihood Function (NLL)

Here we recall some facts about the negative log-likelihood function (NLL) for an exponential family. Let $q_\theta$ denote the density of an exponential family distribution parameterized by $\theta$ and $\ell(\theta; x) := -\log q_\theta(x)$ be its single sample NLL. It can be shown (see e.g. Busa-Fekete et al. [11]) that the derivative and Hessian (also known as *Fisher information*) of the NLL for a single sample $x \sim q_{\theta^\star}$ are given by

$$\nabla_\theta \ell(\theta; x) = \mathbb{E}_{y \sim q_\theta}[T(y)] - T(x), \qquad \nabla_\theta^2 \ell(\theta; x) = \mathrm{Cov}_{y \sim q_\theta}[T(y), T(y)] \succeq 0.$$

Thus the gradient and Hessian of the population NLL $L(\theta) = \mathbb{E}_{x \sim q_{\theta^\star}}[\ell(\theta; x)]$ are as follows:

$$\nabla_\theta L(\theta) = \mathbb{E}_{y \sim q_\theta}[T(y)] - \mathbb{E}_{x \sim q_{\theta^\star}}[T(x)], \qquad \nabla_\theta^2 L(\theta) = \mathrm{Cov}_{y \sim q_\theta}[T(y), T(y)] \succeq 0.$$

As an example, let $q_\theta$ denote the density function of a member of some exponential family and $q_\theta^S$ denote the density truncated to the set $S$. For any $S \subseteq \mathbb{R}^d$, we see that the gradient and Hessian of the *empirical* NLL $\tilde{L}(\theta) = \tilde{L}(\theta; x^{(1)}, \dots, x^{(n)}) := \frac{1}{n} \sum_{i=1}^n -\log q_\theta^S(x^{(i)})$ for $n$ truncated samples have the following form

$$\nabla_\theta \tilde{L}(\theta) = \mathbb{E}_{y \sim q_\theta^S}[T(y)] - \frac{1}{n} \sum_{i=1}^n T(x^{(i)}), \qquad \nabla_\theta^2 L(\theta; x) = \mathrm{Cov}_{y \sim q_\theta}[T(y), T(y)] \succeq 0.$$

Note that under Assumption (S1), both the (untruncated) population and empirical NLL are convex, 1-smooth, and $\lambda$-strongly convex over $\Theta$.

# 3 Privately Estimating Exponential Families via Truncation

Our main result is an efficient truncation-based algorithm for privately learning an exponential family from samples.

**Theorem 3.1.** *Let $\varepsilon, \delta, \alpha, \beta \in (0, 1)$ and suppose the statistical assumptions hold (Assumption 2.2) and computational subroutines exist (Assumption 2.3). There is an SGD-based $(\varepsilon, \delta)$-DP algorithm such that given samples from $q_{\theta^\star}$, outputs an estimate $\hat{\theta}$ satisfying $\|\hat{\theta} - \theta^\star\| \le \alpha$ with probability $1 - \beta$. Moreover, the algorithm has sample complexity $n = \tilde{O}\left( \frac{m \log(1/\eta\beta\delta)}{\lambda^2 \varepsilon} + \frac{m \log(1/\beta)}{\lambda^4 \eta^4 \alpha^2} + \frac{e^{O(1/\lambda^2)} m \log(1/\beta\delta)}{\lambda^2 \alpha \varepsilon} \right)$ and time complexity $\mathrm{poly}(n, m, d)$.*

The exponential dependence on $\lambda$ is an artifact of existing truncated statistics methods [35], which uses a general anti-concentration property of polynomials under log-concave measures in order to control the decay in the strong convexity of the truncated log-likelihood function compared to the untruncated counterpart. Moreover, some dependence on $1/\lambda$ is necessary even when using vanilla SGD to estimate Gaussian parameters. However, for many important exponential families such as Gaussians, this can be mitigated by preconditioning the samples so that $\lambda = \Theta(1)$. We demonstrate how to do this for Gaussians in Section 4.2 (Theorem 4.2).

**Remark 3.2** (Robustness against Existing Truncation). *All of our algorithms only access samples after a preprocessing truncation step. Thus the guarantees of our algorithms all hold if, instead of having sample access to an exponential family, we are only given access to samples which have already undergone truncation to an arbitrary but known survival set.*

**Pseudocode.** Our main DP-SGD subroutine can be found in Algorithm B.1, which we defer to Appendix B.4 due to space constraints. We present a simplified version below for convenience:

1) Truncate samples based on survival set.

2) Preprocess inputs and initialize parameters via warm-start/preconditioning.

3) Minimize *truncated* empirical NLL using DP-SGD (Theorem 3.4)

4) Return (approximate) minimizer as parameter estimate.

## 3.1 Technical Overview

Algorithmically, we first truncate the input dataset to a carefully chosen survival set, which bounds sensitivity but introduces bias for naïve estimators such as the sample mean. Then, we use DP-SGD (Algorithm B.1) to minimize the *truncated* NLL function. This allows us to correct for the bias introduced by the truncation.

For the sake of modularity, we first present our algorithm under the simplifying condition where we are given a $R = O(1)$-distance warm-start and later adapt standard DP estimation tools to obtain $R = O(\frac{\log(1/\rho)}{\lambda})$ in Appendices B.6 and B.7.

**Condition 3.3.** We are given $\theta^{(0)} \in \Theta$ and $\tau^{(0)} = \mathbb{E}_{x \sim q_{\theta^{(0)}}}[T(x)] \in \mathbb{R}^m$ such that $\theta^\star \in B_R(\theta^{(0)})$ for some given $1 \le R = O(1)$.

We emphasize that Condition 3.3 is only stated to simplify our exposition. It is completely removed in Appendices B.6 and B.7 by adapting standard DP Gaussian estimation tools to the truncated exponential family setting.

We use the following analysis of DP-SGD due to Bassily, Smith, and Thakurta [5].[3]

**Theorem 3.4** (Theorem II.1 and Theorem II.4 in [5])**.** *Let $\varepsilon, \delta \in (0,1)$. Suppose $F(w) = \frac{1}{n}\sum_{i=1}^{n} f_i(w)$ is a sum of $\lambda$-strongly convex functions over a closed convex set $K \subseteq \mathbb{R}^m$. Suppose further that we are given stochastic gradient oracles $g_i$ for $\nabla f_i$ satisfying $\|g_i\| \le G$ for all $i \in [n]$. Then there is an $(\varepsilon, \delta)$-DP algorithm that outputs some $\hat{w} \in K$ satisfying*

$$\mathbb{E}[F(\hat{w})] - \min_{w \in K} F(w) \le O\left(\frac{mG^2 \log^2(n/\delta) \log(1/\delta)}{n^2 \lambda \varepsilon^2}\right).$$

*The algorithm runs DP-PSGD for $T = \Theta(n^2)$ iterations with step-size $\frac{1}{\lambda t}$ at iteration $t$ and calls the gradient oracle $T$ times in total.*

We remark that the logarithmic factors in the convergence rate of Theorem 3.4 can be removed under additional assumptions about the condition number of the objective function $F$ [22]. In general, the exponential family distributions we study may be ill-conditioned and we must employ Theorem 3.4 instead.

In the rest of this section, we provide the details for the estimation algorithm of Theorem 3.1 and its proof of correctness. As mentioned, our main workhorse is Theorem 3.4. Sections 3.2 to 3.6 addresses how we satisfy the assumptions of Theorem 3.4 and preprocess the input to attain the desired sample complexity. This is summarized in further detail below.

- Section 3.2 constructs the survival set $S_{\text{SGD}}$ of samples and feasible projection set $K$ of the parameters (Lemma 3.5). Our goal is to ensure that $K$ contains the true parameter and that the truncated NLL function remains strongly convex over $K$.

- For technical reasons, we cannot achieve optimal rates when optimizing the population NLL and must instead optimize the *empirical* NLL. Section 3.3 details this reasoning and proves a uniform convergence property which ensures that the minimizer of the empirical NLL remains a good estimate of the true parameter (Lemma 3.6).

- Section 3.4 addresses how to obtain unbiased estimates of the gradient of the empirical NLL as well as analyzes the norm of the estimates (Lemma 3.7). The latter is a necessary quantity that appears in the DP-SGD analysis (Theorem 3.4).

- Section 3.5 applies Theorem 3.4 to derive the guarantees of our DP-SGD subroutine (Algorithm B.1) and shows how to recover $\theta^\star$ (Lemma 3.8).

---

[3]The original theorem statement is for a deterministic gradient oracle under a Lipschitz condition. It is not hard to see that the same statement holds for an unbiased stochastic gradient oracle with bounded norm.

- Finally, Section 3.6 brings together all the ingredients along with suitable adaptations of standard DP preprocessing tools to prove our main Theorem 3.1.

## 3.2 Strong Convexity (Survival & Projection Sets)

As mentioned in Section 3.1, one sufficient condition for recovering parameters with SGD via Theorem 3.4 is strong convexity. In this section, we specify the truncation operation we impose and show that the truncated NLL is strongly convex over a carefully chosen projection set as long as the survival set has mass $\rho \geq \Omega(1)$.

Let $\theta^{(0)}, \tau^{(0)}$ be as in the simplifying Condition 3.3 and define

$$K := B_{2R}(\theta^{(0)}) \cap \Theta, \qquad S_{\text{SGD}} := \left\{ x \in \mathbb{R}^d : \|T(x) - \tau^{(0)}\| \leq \sqrt{\frac{m}{1-\rho}} + 2R \right\}.$$

**Lemma 3.5.** *Suppose the statistical assumptions (Assumption 2.2)[4] and the simplifying Condition 3.3 hold. Let $\tilde{L}$ denote the empirical NLL over truncated samples with survival set $S_{\text{SGD}}$. Then for any $\theta \in K$, $\boldsymbol{\nabla}^2 \tilde{L}(\theta) \succeq \lambda e^{-O(R^2)} I = \Omega(\lambda) I$.*

The proof of Lemma 3.5 is deferred to Appendix B.1. Crucially, we rely on an anti-concentration inequality due to Carbery and Wright [12] restated by Lee, Wibisono, and Zampetakis [34] (Proposition B.3).

## 3.3 Uniform Convergence of Empirical Likelihood

Having confirmed the strong convexity property necessary to apply Theorem 3.4, we move on to the algorithmic details. Specifically, Theorem 3.4 requires making multiple passes of the data. However, this is problematic as each data point only provides a single unbiased estimate of the gradient of the population NLL. We avoid this complication by instead optimizing the *empirical* NLL, for which each data point provides an unlimited number of unbiased gradient estimates. This requires a uniform convergence type of result to ensure that the empirical minimizer is close to the population minimizer.

**Lemma 3.6.** *Suppose the statistical assumptions (Assumption 2.2) and simplifying Condition 3.3 hold. Let $\tilde{\theta}^\star$ be the minimizer of the $n$-sample empirical NLL for $q_{\theta^\star}^{S_{\text{SGD}}}$ over $K$. Then we have $\|\tilde{\theta}^\star - \theta^\star\|_2 \leq \alpha$ with probability $1 - \beta$ given that $n \geq \Omega(\frac{(m+R^2)\log(1/\beta)}{\lambda^2 \eta^4 \alpha^2})$.*

Lemma 3.6 strengthens prior uniform convergence results [43], which require $\Omega(1/\alpha^4)$ samples, and may be of independent interest. Its proof is deferred to Appendix B.2.

## 3.4 Computing Stochastic Gradients

The previous subsection ensures that we can run DP-SGD for the required number of iterations as per Theorem 3.4. We now address how to compute gradients within each iteration. Similar to previous works on truncated statistics [17], We are able to obtain unbiased stochastic gradients via a simple rejection sampling procedure.

**Lemma 3.7.** *Assume the statistical assumptions (Assumption 2.2) and simplifying Condition 3.3 hold. Fix a sample $x \sim q_{\theta^\star}^{S_{\text{SGD}}}$ and assume we have access to a sampling oracle for $y \sim q_\theta$ (Assumption (C2)). The following holds:*

*(i) There is an an unbiased stochastic gradient estimate $g(\theta)$ for $\boldsymbol{\nabla}\ell(\theta; x)$.*

*(ii) With probability $1 - \beta$, the estimator calls the sampling oracle $O(\log(1/\beta)/\rho)$ times.*

*(iii) The gradient estimate satisfies $\|g(\theta)\|_2 \leq G := O(\sqrt{m} + R)$ with probability $1$.*

The proof of Lemma 3.7 is deferred to Appendix B.3.

---

[4]We only use Assumptions (S1), (S3) and (S4) but state all the statistical assumptions for simplicity of presentation.

## 3.5 DP Empirical Risk Minimization

Now that we are equipped with strong convexity (Lemma 3.5) and bounded gradients (Lemma 3.7) for the necessary number of iterations (Lemma 3.6), we can apply the DP-SGD analysis (Theorem 3.4) to show the following result, whose formal proof is deferred to Appendix B.4.

**Lemma 3.8.** *Let $\varepsilon, \delta, \alpha, \beta \in (0,1)$. Suppose the statistical assumptions hold (Assumption 2.2), the computational subroutines exist (Assumption 2.3), the simplifying Condition 3.3 hold, and that we have sample access to $q_{\theta^\star}$. Let $\tilde{\theta}^\star$ denote the minimizer of the $n$-sample empirical NLL for $q_\theta^{S_{SGD}}$. Algorithm B.1 is an $(\varepsilon, \delta)$-DP algorithm that outputs an estimate $\hat{\theta} \in \Theta$ such that $\mathbb{E}[\|\hat{\theta} - \tilde{\theta}^\star\|^2] \leq \alpha^2$. Moreover, Algorithm B.1 has sample complexity $n = \tilde{O}(\frac{e^{O(R^2)}(m+R\sqrt{m})\log(1/\delta)}{\lambda\alpha\varepsilon})$ and $\mathrm{poly}(m, n, d)$ running time.*

**Remark 3.9** (High-Probability Guarantees; See Corollary B.4). *By using a clustering trick by Daskalakis et al. [17] and taking sufficient samples for uniform convergence to hold (Lemma 3.6), we obtain a high-probability guarantee for estimating the true underlying parameter $\theta^\star$.*

We defer the exact statement of Corollary B.4 and its proof to Appendix B.5.

## 3.6 Proof of Theorem 3.1

In Sections 3.2 to 3.5, we require a constant distance warm start to $\theta^\star$ (Condition 3.3) in order to obtain optimal sample complexity using DP-SGD on the truncated empirical NLL function. This can be removed by adapting standard DP Gaussian warm-start algorithms to truncated exponential families, which we present in Appendices B.6 and B.7 (Lemmas B.5 and B.8). While running the adaptations on truncated data yields biased estimates, such estimates still suffice as a warm start.

Our end-to-end algorithm first obtains a $O(\frac{\log(1/\rho)}{\lambda})$-distance warm-start via Lemmas B.5 and B.8. Then, we use DP-SGD to address the bias introduced by the truncation involved in the rough estimations (Corollary B.4). This yields a proof of Theorem 3.1 with the desired sample and time complexity $\tilde{O}(\frac{m\log(1/\eta\beta\delta)}{\lambda^2\varepsilon} + \frac{m\log(1/\beta)}{\lambda^4\eta^4\alpha^2} + \frac{e^{O(1/\lambda^2)}m\log(1/\beta\delta)}{\lambda^2\alpha\varepsilon})$.

# 4 Private Gaussian Estimation

In order to contextualize our results, we instantiate our general algorithm from Theorem 3.1 for the well-studied case of Gaussian estimation and demonstrate that we can recover the known optimal sample complexities up to logarithmic factors. We emphasize that the first polynomial-time algorithms with optimal sample complexity were achieved by Hopkins, Kamath, and Majid [26] and Hopkins et al. [27], and this section demonstrates that we can recover the optimal sample complexities with our more general algorithmic framework.

## 4.1 Private Gaussian Mean Estimation

We first study Gaussian mean estimation. That is, there is an underlying $d$-dimensional data-generating distribution $\mathcal{N}(\mu^\star, I)$ to which we have sample access. We would like to privately estimate $\mu^\star$.

In Appendix C.1, we verify that the statistical and computational assumptions (Assumptions 2.2 and 2.3) hold for Gaussian mean estimation, leading to the following corollary of Theorem 3.1.

**Theorem 4.1.** *Let $\varepsilon, \delta, \alpha, \beta \in (0,1)$. There is an $(\varepsilon, \delta)$-DP algorithm such that given samples from a Gaussian distribution $\mathcal{N}(\mu^\star, I)$, outputs an estimate $\hat{\mu}$ satisfying $\|\hat{\mu} - \mu^\star\| \leq \alpha$ with probability $1 - \beta$. Moreover, the algorithm has sample complexity $n = \tilde{O}(\frac{d\log(1/\beta\delta)}{\varepsilon} + \frac{d\log(1/\beta)}{\alpha^2} + \frac{d\log(1/\beta\delta)}{\alpha\varepsilon})$ and running time $\mathrm{poly}(n, d)$.*

## 4.2 Private Gaussian Covariance Estimation

We next specialize our algorithm to the case of Gaussian covariance estimation. That is, there is an underlying $d$-dimensional data-generating distribution $\mathcal{N}(0, \Sigma^\star)$ to which we have sample access. We would like to estimate $\Sigma^\star \in \mathbb{S}_+^d$ under differential privacy constraints, where $\mathbb{S}_+^d$ denotes the space of $d \times d$ positive-definite matrices.

**Theorem 4.2.** *Let* $\varepsilon, \delta, \alpha, \beta \in (0,1)$ *and suppose that* $\lambda I \preceq \Sigma^\star \preceq \Lambda I$. *There is an* $(\varepsilon, \delta)$-*DP algorithm such that given samples from a Gaussian distribution* $\mathcal{N}(0, \Sigma^\star)$, *outputs an estimate* $\widehat{\Sigma}$ *satisfying* $\|I - (\Sigma^\star)^{-\frac{1}{2}} \widehat{\Sigma} (\Sigma^\star)^{-\frac{1}{2}}\|_F \leq \alpha$ *with probability* $1 - \beta$. *Moreover, the algorithm has sample complexity* $n = \tilde{O}\left(\frac{d^{1.5} \log(\Lambda/\lambda\beta\delta)}{\varepsilon} + \frac{d^2 \log(1/\beta\delta)}{\varepsilon} + \frac{d^2 \log(1/\beta)}{\alpha^2} + \frac{d^2 \log(1/\beta\delta)}{\alpha\varepsilon}\right)$ *and running time* $\text{poly}(n, d)$.

We present the proof of Theorem 4.2 in Appendix C.2. In addition to verifying the statistical and computational assumptions, we also need a private preconditioning algorithm in order to avoid polynomial dependence on the condition number $\Lambda/\lambda$. We present a preconditioner adapted from the work of Biswas et al. [7] in Appendix C.4. The adapted algorithm essentially estimates the covariance of the *truncated* Gaussian distribution, which will be a biased estimate of original covariance but suffices to precondition the samples.

## 5 Impact & Future Work

We introduce a novel paradigm of private algorithm design through truncation and demonstrate its versatility by designing the first efficient algorithm for estimating unbounded high-dimensional exponential families. We further demonstrate its sample efficiency by recovering the optimal sample complexities for standard private statistical tasks such as Gaussian mean and covariance estimation. Our methods may enable more practical and scalable deployment of privacy-preserving data analysis tools in settings where extreme data values are common but privacy is critical.

It would be interesting to see further applications of truncated statistic techniques in private algorithm design, such as regression [15, 16, 18] and linear dynamics [41].

## Acknowledgements

We thank Alkis Kalavasis for the insightful discussions. Felix Zhou acknowledges the support of the Natural Sciences and Engineering Research Council of Canada (NSERC).

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

# A    Deferred Preliminaries

## A.1    Differential Privacy

We first recall the following preliminaries from differential privacy.

**Definition A.1** (Differential Privacy; Dwork et al. [20])**.** Given $\varepsilon > 0$ and $\delta \in (0, 1)$, a randomized algorithm $\mathcal{A} : \mathcal{X}^n \to \mathcal{Y}$ is $(\varepsilon, \delta)$-DP if, for every pair of neighboring datasets $D, D' \in \mathcal{X}^n$ that differ by a single entry (i.e. neighboring datasets) and for all subsets $U$ of the output space,

$$\Pr[\mathcal{A}(D) \in U] \leq e^\varepsilon \cdot \Pr[\mathcal{A}(D') \in U] + \delta \,.$$

A fundamental tool in designing DP algorithms is the Gaussian mechanism. We write $D \sim D'$ to denote two neighboring datasets.

**Proposition A.2** (Gaussian Mechanism; Dwork and Roth [19])**.** *Let $f : \mathcal{X}^n \to \mathbb{R}^d$ be an arbitrary function $d$-dimensional with $\ell_2$-sensitivity $\Delta_2(f) := \max_{D \sim D'} \|f(D) - f(D')\|$. For any $\varepsilon, \delta \in (0, 1)$, the mechanism that outputs $f(D) + \xi$ where $\xi \sim \mathcal{N}(0, \sigma^2 I)$ is $(\varepsilon, \delta)$-DP for*

$$\sigma \geq \frac{2\Delta_2(f) \ln(1.25/\delta)}{\varepsilon} \,.$$

An important property of differential privacy is that performing computation on a privatized output cannot lose additional privacy:

**Theorem A.3** (Post-Processing; Dwork and Roth [19])**.** *Let $\mathcal{M}$ be an $(\varepsilon, \delta)$-DP mechanism and $g$ be any arbitrary random mapping. Then $g(\mathcal{M}(\cdot))$ is $(\varepsilon, \delta)$-differentially private.*

Moreover, multiple computations on a dataset incur privacy cost in a natural manner.

**Theorem A.4** (Simple Composition; Dwork and Roth [19])**.** *Let $\mathcal{M}_1$ and $\mathcal{M}_2$ be $(\varepsilon_1, \delta_1)$ and $(\varepsilon_2, \delta_2)$-DP mechanisms, respectively. Then the (adaptive) composition $\mathcal{M}_2(\cdot, \mathcal{M}_1(\cdot))$ is $(\varepsilon_1 + \varepsilon_2, \delta_1 + \delta_2)$-DP.*

On the other hand, executing a private mechanism on disjoint partitions of the same dataset does not incur any additional privacy cost.

**Theorem A.5** (Parallel composition of differential privacy; McSherry [38])**.** *Let $\mathcal{M}$ be an $(\varepsilon, \delta)$-DP mechanism and $D_1, \ldots, D_k$ be $k$ disjoint subsets of the dataset $D$. Then the mechanism that outputs $(\mathcal{M}(D_1), \ldots, \mathcal{M}(D_k))$ is $(\varepsilon, \delta)$-DP.*

## A.2    Concentration Inequalities

**Theorem A.6** (Lemma 1 in Laurent and Massart [33])**.** *Let $Z \sim \mathcal{N}(0, I)$. Then with probability $1 - \beta$,*

$$\|Z\|_2^2 \leq d + \sqrt{2d \log(1/\beta)} + 2 \log(1/\beta) \,.$$

Recall a centered random variable $X$ is said to be $(\nu^2, 1/\eta)$-*subexponential* if

$$\mathbb{E}[e^{\lambda X}] \leq \exp\left(\frac{\nu^2 \lambda^2}{2}\right)$$

for all $|\lambda| \in (0, \eta)$. If $X_i$ is $(\nu_i^2, 1/\eta_i)$-subexponential for $i \in [n]$, then it is well-known [48, Section 2.1.3] that its sum $\sum_i X_i$ is $(\nu^2, 1/\eta)$-subexponential for

$$\nu := \sqrt{\sum_i \nu_i^2}, \qquad \frac{1}{\eta} := \max_i \frac{1}{\eta_i} \,.$$

Subexponential variables enjoy the following concentration properties.

**Proposition A.7** (Proposition 2.9 in [48])**.** *For a $(\nu^2, 1/\eta)$-subexponential variable $X$,*

$$\Pr[X \geq t] \leq \begin{cases} e^{-\frac{t^2}{2\nu^2}}, & t \in (0, \nu^2 \eta), \\ e^{-\frac{t\eta}{2}}, & t \geq \nu^2 \eta. \end{cases}$$

*and*

$$\Pr[X \leq -t] \leq \begin{cases} e^{-\frac{t^2}{2\nu^2}}, & t \in (0, \nu^2 \eta), \\ e^{-\frac{t\eta}{2}}, & t \geq \nu^2 \eta. \end{cases}$$

A specific example is the subexponentiality of the sufficient statistics of exponential families.

**Proposition A.8** (Claim 1 in [34])**.** *Suppose Assumptions (S1) and (S2) hold. Then for any unit vector $u \in \mathbb{R}^m$ and $x \sim q_{\theta^\star}$, $u^\top (\mathbb{E}_{y \sim q_{\theta^\star}}[T(y)] - T(x))$ is $(1, 1/\eta)$-subexponential (cf. Appendix A.2).*

A useful concentration result for bounded vectors is the following.

**Theorem A.9** (Vector Bernstein Inequality; Lemma 18 in [31])**.** *Let $X^{(1)}, \ldots, X^{(n)}$ be independent random vectors with common dimension $d$ satisfying the following for all $i \in [n]$:*

    *(i) $\mathbb{E}[X^{(i)}] = 0$*

    *(ii) $\|X^{(i)}\| \leq R$*

    *(iii) $\mathbb{E}[\|X^{(i)}\|^2] \leq G^2$*

*Let $X := \frac{1}{n} \sum_{i=1}^{n} X^{(i)}$. Then for any $\alpha \in (0, G^2/R)$,*

$$\Pr[\|X\| \geq \alpha] \leq \exp\left(-\frac{\alpha^2 n}{8G^2} + \frac{1}{4}\right).$$

For covariance estimation, we also require the following spectral concentration bound for symmetric Gaussian matrices.

**Theorem A.10** (Corollary 2.3.6 in Tao [46])**.** *Let $Y$ be a random $d \times d$ symmetric matrix $Y$ with $Y_{ij} \sim \mathcal{N}(0, \sigma^2)$. For $d$ sufficiently large, there are absolute constants $C, c > 0$ such that for all $t \geq C$,*

$$\Pr\left[\|Y\|_2 > t\sigma\sqrt{d}\right] \leq C \exp(-ctd).$$

### A.3 Omitted Proofs from Section 2.2

We now restate and prove the guarantees of our preprocessing algorithm.

**Lemma 2.1.** *Fix $n \geq 1$ and let $N \in \mathbb{N}$. Let $D \in \mathbb{R}^{d \times N}$ be an $N$-sample dataset, $D' \in \mathbb{R}^{d \times N}$ be obtained from $D$ by modifying a single entry, and $S \subseteq \mathbb{R}^d$. Write $D_S, D'_S$ to denote the datasets obtained from $D, D'$ by discarding entries that fall outside of $S$. There is a preprocessing algorithm $\mathcal{A}$ such that*

    *(i) $\mathcal{A}(D'_S) \in S^n$ can be obtained from $\mathcal{A}(D_S) \in S^n$ by modifying a single element,*

    *(ii) for any $(\varepsilon, \delta)$-DP algorithm $\mathcal{B}$ with respect to neighboring truncated datasets, the composition $\mathcal{B}(\mathcal{A}(D_S))$ is $(\varepsilon, \delta)$-DP with respect to neighboring (untruncated) datasets $D, D'$, and*

    *(iii) if $D$ is sampled i.i.d. from some distribution $p$ such that $p(S) =: \rho$, then $\mathcal{A}(D)$ contains i.i.d. samples from the truncated distribution $p^S$ with probability $1 - \beta$, provided $N = \Omega(\frac{n \log(1/\beta)}{\rho})$.*

**Pseudocode.** See Line 1 to Line 4 of Algorithm B.1.

*Proof.* We wish to produce two neighboring datasets of size $n$ by discarding points that lie outside of the survival set.

Proof of (i): We first analyze the relationship between neighboring datasets after the initial truncation. Suppose $D'$ is obtained from $D$ by modifying $x \in D$ to $y \in D'$, then $D_S, D'_S$ fall into one of three cases:

    I) If $x, y \notin S$, then $D_S = D'_S$.

    II) If $x, y \in S$, then $D'_S$ can be obtained from $D_S$ by modifying $x$ to $y$, i.e. they are again neighboring datasets.

    III) If $|S \cap \{x, y\}| = 1$, then $D'_S$ can be obtained from $D_S$ by adding or deleting an element.

We focus on the more challenging Case III). Without loss of generality, suppose that $x \notin S$ and $y \in S$ so that $D'_S$ is obtained from $D_S$ by adding $y$. If $|D_S| < |D'_S| \leq n$, $\mathcal{A}$ picks any data-independent $x_{dummy} \in S$ and add copies of $x_{dummy}$ to the truncated dataset until there are $n$ elements. Then $\mathcal{A}(D), \mathcal{A}(D')$ are neighboring $n$-sample datasets with elements from $S$. Otherwise, if $n \leq |D_S| < |D'_S|$, $\mathcal{A}$ keeps the first $n$ datapoints of the truncated dataset so that there are again $n$ elements. Then $\mathcal{A}(D), \mathcal{A}(D')$ are once again neighboring $n$-sample datasets with elements from $S$, regardless of which entry of the dataset differs between $D, D'$. Note that in either scenarios, we need to enforce the dataset size by adding or deleting elements, but never both.

Cases I), II) are more clear as they are already neighboring datasets and the additional processing does not change this fact.

Proof of (ii): Next, we analyze the privacy guarantees of the composition $\mathcal{B}(\mathcal{A}(D_S))$. This essentially follows from (i), as $\mathcal{A}(D_S), \mathcal{A}(D'_S)$ can be obtained from each other as sets by modifying one element. The only subtle difference is that in case III) above, prior to the shuffling step, the order of entries is possibly not "aligned", i.e. there could be more than one index $i \in [n]$ where $\mathcal{A}(D_S)_i \neq \mathcal{A}(D'_S)_i$. The extra shuffling step in Line 4 of Algorithm B.1 ensures that under a suitable coupling of $\mathcal{A}(D_S), \mathcal{A}(D'_S)$, they are always neighboring datasets, i.e. there is only a single index $i^\star$ such that $\mathcal{A}(D_S)_{i^\star} \neq \mathcal{A}(D'_S)_{i^\star}$. We can take this coupling to be the one where every element in $\mathcal{A}(D_S) \cap \mathcal{A}(D'_S)$ is always shuffled to the same index. Then the privacy of $\mathcal{B}(\mathcal{A}(D_S))$ follows by the privacy guarantees of $\mathcal{B}$.

Proof of (iii): We finish by analyzing the utility guarantees. By a concentration inequality for sums of geometric random variables [28, Corollary 2.4], for $N = \Omega(\frac{n \log(1/\beta)}{\rho})$, it holds with probability $1 - \beta$ that there are at least $n$ samples that remain after the preprocessing step. Conditioned on this, the preprocessing simply deletes some possibly additional samples from $p^S$ but does not add dummy points and the output is as desired. $\qquad\square$

# B  Omitted Proofs from Section 3

## B.1  Proof of Lemma 3.5

We now restate and prove Lemma 3.5.

**Lemma 3.5.** *Suppose the statistical assumptions (Assumption 2.2)[5] and the simplifying Condition 3.3 hold. Let $\tilde{L}$ denote the empirical NLL over truncated samples with survival set $S_{SGD}$. Then for any $\theta \in K$, $\nabla^2 \tilde{L}(\theta) \succeq \lambda e^{-O(R^2)} I = \Omega(\lambda) I$.*

The proof relies on the following facts from the truncated statistics literature.

**Proposition B.1.** *Suppose that Assumption (S1) and Condition 3.3 hold. Then for any $\theta \in K$, $q_\theta(S_{SGD}) \geq \rho = \Omega(1)$.*

*Proof of Proposition B.1.* From elementary probability,
$$\mathbb{E}_{x \sim q_\theta}\left[\|T(x) - \mathbb{E}_{y \sim q_\theta}[T(y)]\|^2\right] = \mathrm{tr}\left(\mathrm{Cov}_{x \sim q_\theta}[T(x), T(x)]\right) \leq m.$$
Then $q_\theta$ puts $\rho$ mass on the set
$$S_\theta := \left\{ x : \|T(x) - \mathbb{E}_{y \sim q_\theta}[T(y)]\| \leq \sqrt{\frac{m}{1 - \rho}} \right\}.$$
Let $f$ denote the population NLL function with respect to $\theta^{(0)}$ so that $\nabla f(\theta^{(0)}) = 0$. By Assumption (S1), $\nabla^2 f \preceq I$ over $K$. Thus for any $\theta \in B_R(\theta^{(0)}) \cap \Theta$,
$$\|\mathbb{E}_{y \sim q_\theta}[T(y)] - \mathbb{E}_{x \sim q_{\theta^{(0)}}}[T(x)]\| = \|\nabla f(\theta)\|$$
$$= \|\nabla f(\theta) - \nabla f(\theta^{(0)})\|$$
$$\leq \|\theta - \theta^{(0)}\| \qquad \text{(By 1-smoothness of } f)$$
$$\leq R.$$
This ensures that $S_\theta \subseteq S_{SGD}$ so that $q_\theta(S_{SGD}) \geq \rho = \Omega(1)$. $\qquad\square$

---

[5]We only use Assumptions (S1), (S3) and (S4) but state all the statistical assumptions for simplicity of presentation.

**Proposition B.2** (Lemma 3.4 in [34]). *Suppose Assumption (S1) holds. Then for any $\theta, \theta' \in \Theta$ and $S \subseteq \mathbb{R}^d$, $q_{\theta'}(S) \geq q_{\theta'}(S)^2 \cdot \exp\left(-\frac{3}{2}\|\theta - \theta'\|^2\right)$.*

**Proposition B.3** (Lemma 3.2 in [34]). *Fix $\theta \in \Theta$. Suppose Assumptions (S1), (S3) and (S4) holds and $S \subseteq \mathbb{R}^d$ satisfies $q_\theta(S) > 0$. Then $\mathrm{Cov}_{y \sim q_\theta^S}[T(y), T(y)] \succeq \frac{1}{2}\left(\frac{q_\theta(S)}{4Ck}\right)^{2k}\lambda I$, where $C > 0$ is some absolute constant.*

We are now ready to prove Lemma 3.5.

*Proof of Lemma 3.5.* By Condition 3.3, $K$ certainly contains $\theta^\star$. In particular, Proposition B.1 ensures that we have $q_{\theta^\star}(S_{\mathrm{SGD}}) \geq \rho = \Omega(1)$. An application Proposition B.2 allows us to deduce that $q_\theta$ puts $\Omega(q_{\theta^\star}(S_{\mathrm{SGD}})^2) = \Omega(\rho^2)$ mass on $S_{\mathrm{SGD}}$ for every $\theta \in K$. But then by Proposition B.3,

$$\mathrm{Cov}_{y \sim q_\theta^{S_{\mathrm{SGD}}}}[T(y), T(y)] \succeq \frac{1}{2}\left(\frac{\rho^2 e^{-6R^2}}{4Ck}\right)^{2k}\lambda I \text{, concluding the proof.} \qquad \square$$

## B.2 Proof of Lemma 3.6

We now prove Lemma 3.6, whose statement is copied below for convenience.

**Lemma 3.6.** *Suppose the statistical assumptions (Assumption 2.2) and simplifying Condition 3.3 hold. Let $\tilde{\theta}^\star$ be the minimizer of the $n$-sample empirical NLL for $q_{\theta^\star}^{S_{\mathrm{SGD}}}$ over $K$. Then we have $\|\tilde{\theta}^\star - \theta^\star\|_2 \leq \alpha$ with probability $1 - \beta$ given that $n \geq \Omega\left(\frac{(m+R^2)\log(1/\beta)}{\lambda^2\eta^4\alpha^2}\right)$.*

*Proof.* The sufficient statistics of $q_{\theta^\star}^{S_{\mathrm{SGD}}}$ has radius $r = O(\sqrt{m} + R)$ by construction. We can thus apply a vector Bernstein inequality (Theorem A.9) to see that for any $\alpha \in (0, 1)$,

$$\Pr\left[\left\|\mathbb{E}_{y \sim q_{\theta^\star}}[T(y)] - \frac{1}{n}\sum_{i=1}^n T(x^{(i)})\right\| \geq \alpha\right] \leq \exp\left(-\frac{\alpha^2 n}{8r^2} + \frac{1}{4}\right).$$

Let $c > 0$ be the constant guaranteed by Lemma 3.5 such that $\tilde{L}$ is $c\lambda$-strongly convex over $K$. We have $\|\nabla\tilde{L}(\theta^\star)\| \leq c\lambda\eta^2/4r$ with probability $1 - \beta/2$ given that

$$n \geq \Omega\left(\frac{(m + R^2)\log(1/\beta)}{\lambda^2\eta^4}\right).$$

But then by strong convexity,

$$\frac{c\lambda}{2}\|\tilde{\theta}^\star - \theta^\star\|^2 \leq \underbrace{\tilde{L}(\tilde{\theta}^\star) - \tilde{L}(\theta^\star)}_{\leq 0} + \langle\nabla\tilde{L}(\theta^\star), \theta^\star - \tilde{\theta}^\star\rangle \leq \frac{c\lambda\eta^2}{4r}\cdot 2r.$$

Thus $\|\tilde{\theta}^\star - \theta^\star\| \leq \eta$. By an application of the triangle inequality, this ensures that $\|\tilde{\theta}^\star - \theta^{(0)}\| \leq R + \eta \leq 2R$ so that $\tilde{\theta}^\star \in K$.

Conditioned on $\tilde{\theta}^\star \in K$ and using the fact that $\tilde{L}$ is strongly convex over $K$, we see that $\tilde{L}$ in fact satisfies a $c\lambda$-PL inequality:

$$\frac{1}{2c\lambda}\|\nabla\tilde{L}(\theta^\star)\|^2 \geq \tilde{L}(\theta^\star) - \tilde{L}(\tilde{\theta}^\star) \geq \frac{c\lambda}{2}\|\theta^\star - \tilde{\theta}^\star\|^2.$$

We have $\|\nabla\tilde{L}(\theta^\star)\| \leq c\lambda\alpha$ with probability $1 - \beta/2$ provided that

$$n \geq \Omega\left(\frac{(m + R^2)\log(1/\beta)}{\lambda^2\alpha^2}\right).$$

In particular, $\|\theta^\star - \tilde{\theta}^\star\| \leq \alpha$. This concludes the proof. $\qquad \square$

### B.3 Proof of Lemma 3.7

We now restate and prove Lemma 3.7.

**Lemma 3.7.** *Assume the statistical assumptions (Assumption 2.2) and simplifying Condition 3.3 hold. Fix a sample $x \sim q_{\theta^\star}^{S_{SGD}}$ and assume we have access to a sampling oracle for $y \sim q_\theta$ (Assumption (C2)). The following holds:*

    *(i) There is an an unbiased stochastic gradient estimate $g(\theta)$ for $\boldsymbol{\nabla}\ell(\theta; x)$.*

    *(ii) With probability $1 - \beta$, the estimator calls the sampling oracle $O(\log(1/\beta)/\rho)$ times.*

    *(iii) The gradient estimate satisfies $\|g(\theta)\|_2 \leq G := O(\sqrt{m} + R)$ with probability 1.*

*Proof.* Fix a sample $x \sim q_{\theta^\star}^{S_{SGD}}$. Given a sampling oracle to $q_\theta$, we can perform rejection sampling to obtain $y \sim q_\theta^{S_{SGD}}$. Then we have stochastic access to $\boldsymbol{\nabla}_\theta \tilde{L}$ given by $g(\theta) := T(y) - T(x)$. Moreover, $\|T(x) - T(y)\| \leq \|T(x) - \tau^{(0)}\| + \|T(x) - \tau^{(0)}\| \leq O(\sqrt{m} + R)$ by the choice of $S_{SGD}$. Hence we have a deterministic bound $\|g(\theta)\| \leq G := O(\sqrt{m} + R)$ on the norm of the stochastic gradient. $\square$

### B.4 Proof of Lemma 3.8

we now restate and prove Lemma 3.8.

**Lemma 3.8.** *Let $\varepsilon, \delta, \alpha, \beta \in (0, 1)$. Suppose the statistical assumptions hold (Assumption 2.2), the computational subroutines exist (Assumption 2.3), the simplifying Condition 3.3 hold, and that we have sample access to $q_{\theta^\star}$. Let $\tilde{\theta}^\star$ denote the minimizer of the $n$-sample empirical NLL for $q_\theta^{S_{SGD}}$. Algorithm B.1 is an $(\varepsilon, \delta)$-DP algorithm that outputs an estimate $\hat{\theta} \in \Theta$ such that $\mathbb{E}[\|\hat{\theta} - \tilde{\theta}^\star\|^2] \leq \alpha^2$. Moreover, Algorithm B.1 has sample complexity $n = \tilde{O}\left(\frac{e^{O(R^2)}(m+R\sqrt{m})\log(1/\delta)}{\lambda\alpha\varepsilon}\right)$ and $\mathrm{poly}(m, n, d)$ running time.*

**Pseudocode.** See Algorithm B.1 for the pseudocode. We note that the main difference from standard applications of DP-SGD is the initial truncation step which discards samples that fall outside of the survival set $S$. This provides an easy bound on the sensitivity of the gradient, but requires optimizing the *truncated* NLL as opposed to the regular NLL in order to address the bias introduced by the initial truncation step.

**Analysis.** Applying Theorem 3.4 yields a proof of Lemma 3.8.

*Proof of Lemma 3.8.* We know that $\tilde{L} : K \to \mathbb{R}$ is $\lambda e^{-O(R^2)}$-strongly convex by Lemma 3.5. Moreover, Lemma 3.7 guarantees that we have stochastic gradients with bounded norm $G = O(\sqrt{m} + R)$. Let $\hat{\theta}$ be the output of Theorem 3.4. We see that it satisfies

$$\mathbb{E}[\|\hat{\theta} - \tilde{\theta}^\star\|^2] \leq O\left(\frac{e^{O(R^2)}(m^2 + mR^2)\log^2(n/\delta)\log(1/\delta)}{\lambda^2 n^2 \varepsilon^2}\right).$$

Thus in order to reduce the expected squared distance to $(\alpha/16)^2$, it suffices to take

$$n \geq \Omega\left(\frac{e^{O(R^2)}(m + R\sqrt{m})\log(n/\delta)\sqrt{\log(1/\delta)}}{\lambda\alpha\varepsilon}\right).$$

This concludes the proof. $\square$

### B.5 Proof of High-Probability Estimation

Here, we state and prove the high-probability version of Lemma 3.8.

**Corollary B.4.** *Let $\varepsilon, \delta, \alpha, \beta \in (0, 1)$. Suppose Assumption 2.2, Assumption 2.3, and Condition 3.3 hold and that we have sample access to $q_{\theta^\star}^{S_{SGD}}$. There is an $(\varepsilon, \delta)$-DP algorithm that outputs an*

**Algorithm B.1:** DP-SGD with Truncation

---

**Input:** $N$-sample dataset $D$, desired truncated samples $n$, privacy parameters $\varepsilon, \delta \in (0,1)$, survival set $S$, truncated sensitivity $\Delta > 0$, warm-start $\theta^{(0)} \in \mathbb{R}^m$, accuracy $\alpha \in (0,1)$, step-size function $\gamma(t) : \mathbb{Z}_+ \to \mathbb{R}_+$, projection set $K$

**Output:** estimator $\hat{\theta}$ for $\theta^\star$

---

1   $D_S \leftarrow \{x \in D : x \in S\}$

2   $D_S \leftarrow D_S \cup \left\{ x_{dummy}^{(i)} : i \in [n - |D_S|] \right\}$   ▷ Fill with dummy elements if there are less than $n$ elements

3   $D_S \leftarrow D_S[1:n]$           ▷ Only keep first $n$ elements if there are more than $n$

4   Shuffle $D_S$ uniformly at random

5   $\sigma^2 \leftarrow \frac{32\Delta^2 \log(n/\delta) \log(1/\delta)}{\varepsilon^2}$   ▷ Taken from Bassily, Smith, and Thakurta [5] (Theorem 3.4)

6   **for** *iteration* $t = 1, \ldots, n^2$ **do**

7      $x \sim D_S$ sampled with replacement

8      $y \sim q_{\theta^{(t-1)}}^S$          ▷ Rejection sampling using sampling oracle (Assumption (C2))

9      $g^{(t)} \leftarrow T(x) - T(y)$          ▷ Gradient computation (Lemma 3.7)

10     $\xi \sim \mathcal{N}(0, \sigma^2 I)$

11     $\theta^{(t)} \leftarrow \theta^{(t-1)} - \gamma(t)[g^{(t)} + \xi]$

12     $\theta^{(t)} \leftarrow \text{proj}_K(\theta^{(t)})$

13   **return** $\theta^{(n^2)}$

---

estimate $\hat{\theta} \in \Theta$ such that $\|\hat{\theta} - \theta^\star\|^2 \leq \alpha^2$ with probability $1 - \beta$. Moreover, the algorithm has sample complexity $n = \tilde{O}\left( \frac{(m+R^2)\log(1/\beta)}{\lambda^2 \eta^4 \alpha^2} + \frac{e^{O(R^2)}(m+R\sqrt{m})\log(1/\beta\delta)}{\lambda\alpha\varepsilon} \right)$ and $\text{poly}(m, d, n)$ running time.

*Proof.* Similar to Daskalakis et al. [17], we perform a boosting trick. Consider the output $\tilde{\theta}$ of a single execution of Algorithm B.1. By Lemma 3.8 and Markov's inequality, $\|\tilde{\theta} - \theta^\star\|^2 \geq \alpha/4$ with probability at most $1/4$. By a multiplicative Chernoff bound, repeating $v = O(\log(1/\beta))$ independent executions of Algorithm B.1 ensures that at least $2/3$ of the outputs $\tilde{\theta}^{(1)}, \ldots, \tilde{\theta}^{(v)}$ are $\alpha/4$-close to $\tilde{\theta}^\star$. Now, any of the $2/3$ outputs are within $\alpha/2$ distance to each other, Hence by outputting any of the $v$ points, say $\hat{\theta}$, that is within $\alpha/2$ distance with at least $v/2$ of the other points ensures that it is within a distance of $\alpha$ from $\tilde{\theta}^\star$ with probability $1 - \beta$.

Furthermore, by Lemma 3.6, we have that $\|\tilde{\theta}^\star - \theta^\star\| \leq \alpha$ given that

$$n \geq \Omega\left( \frac{(m + R^2)\log(1/\beta)}{\lambda^2 \eta^4 \alpha^2} \right).$$

Boosting increases the sample complexity by a factor of $O(\log(1/\beta))$. Note that there is no additional privacy loss since we can think of the algorithm as running on $v$ disjoint chunks of the dataset (Theorem A.5). Thus we require

$$n \geq \tilde{\Omega}\left( \frac{(m + R^2)\log(1/\beta)}{\lambda^2 \eta^4 \alpha^2} + \frac{e^{O(R^2)}(m + R\sqrt{m})\log(n/\delta)\sqrt{\log(1/\delta)}\log(1/\beta)}{\lambda\alpha\varepsilon} \right).$$

This concludes the proof. $\qquad\qquad\qquad\qquad\qquad\qquad\qquad\qquad\qquad\qquad\qquad\qquad\qquad$ $\square$

## B.6   Recursive Warm-Start

We now present a simple recursive method that obtains a $O(\frac{\log(1/\rho)}{\lambda})$-distance warm start with logarithmic dependence on the prior radius $R$. The algorithm is adapted from the work of Biswas et al. [7] for Gaussians. Similar to the rest of our work, our algorithm only requires truncated sample access to an exponential family and thus generalizes the work of Biswas et al. [7] from Gaussians

to truncated exponential families. In Appendix B.7, we will see how to obtain a $\text{poly}(m)$-distance warm-start without any prior, thus completely removing the dependence on $R$.

**Lemma B.5.** *Suppose that the exponential family has bounded covariances (Assumption (S1)) and that we have access to a moment matching oracle (Assumption (C3)). Algorithm B.2 is an $(\varepsilon, \delta)$-DP algorithm such that given samples from $q_{\theta^\star}$ and a prior parameter $\theta^{(0)}$ such that $\|\theta^{(0)} - \theta^\star\| \leq R$, outputs some $\hat{\theta}$ and $\hat{\tau} = \mathbb{E}_{x \sim p_{\hat{\theta}}}[T(x)]$ such that $\|\hat{\theta} - \theta^\star\| \leq O(\frac{\log(1/\rho)}{\lambda})$ with probability $1 - \beta$. Moreover, the algorithm has sample complexity $n = \tilde{O}\left(\frac{m \log(R/\beta)}{\lambda^2} + \frac{m \log(R/\beta\delta)}{\lambda\varepsilon}\right)$ and running time $\text{poly}(m, n, d)$.*

The proof of the result above is deferred to Appendix B.6. We emphasize Lemma B.5 requires only truncated sample access for an exponential family. While there are private warm-start algorithms for Gaussians, it is not clear if their guarantees hold for truncated exponential families.

**Pseudocode.** See Algorithm B.2 for pseudocode of the warm-start algorithm. We remark that it is a straightforward adaptation of the Gaussian estimation algorithm from Biswas et al. [7] to the case of truncated exponential families. However, the adaptation only provides a constant-distance warm start since the initial truncation step introduces bias.

---

**Algorithm B.2:** Recursive Warm-Start

**Input:** dataset $D$, number of desired samples $n$, privacy parameters $\varepsilon, \delta \in (0, 1)$, initial prior $\theta^{(0)} \in \mathbb{R}^m$, initial expected sufficient statistic $\tau^{(0)} = \mathbb{E}_{x \sim q_{\theta^{(0)}}}[T(x)]$, initial distance $R > 0$, survival probability $\rho \in (0, 1)$

**Output:** estimator $\hat{\theta}$ for $\theta^\star$ and $\hat{\tau} = \mathbb{E}_{x \sim \hat{\theta}}[T(x)]$

1   $\varepsilon' \leftarrow \frac{\varepsilon}{\log(R/\sqrt{m})+1}$

2   $\delta' \leftarrow \frac{\delta}{\log(R/\sqrt{m})+1}$

3   $\sigma \leftarrow O\left(\frac{(R+\sqrt{m})\log(1/\delta')}{n\varepsilon'}\right)$

4   **for** $i = 0, \ldots, v = \log(R/\sqrt{m})$ **do**

5      $S_{\text{warm}, 2^{-i}R} \leftarrow \{x \in \mathbb{R}^d : \|T(x) - \tau^{(i)}\| \leq \frac{\sqrt{m}}{1-\rho} + 2^{-i}R\}$

6      Produce $n$ truncated samples $x^{(1)}, \ldots, x^{(n)}$ from $D_{S_{\text{warm}, 2^{-i}R}}$      ▷ (Lemma 2.1)

7      $\tau \leftarrow \frac{1}{n} \sum_{j=1}^n T(x^{(j)})$          ▷ sensitivity $\Delta = O(\frac{2^{-i}R+\sqrt{m}}{n})$

8      $\xi \sim \mathcal{N}(0, \sigma^2 I)$

9      $\tau^{(i+1)} \leftarrow \tau + \xi$        ▷ Gaussian mechanism (Proposition A.2)

10   $\theta^{(v+1)} \leftarrow \text{MomentMatch}(\tau^{(v+1)})$        ▷ (Assumption (C3))

11   **return** $\theta^{(v+1)}, \tau^{(v+1)}$

---

**Analysis.** Once again, the idea is to impose a truncation about the samples so that we can work with a bounded random variable. However, we need to ensure that the survival set is chosen to have constant mass. Similar to Section 3.2, we consider the survival set

$$S_{\text{warm}, 2^{-i}R} \leftarrow \left\{x \in \mathbb{R}^d : \|T(x) - \tau^{(i)}\| \leq \frac{\sqrt{m}}{1-\rho} + 2^{-i}R\right\},$$

where we iteratively shrink the prior distance with each iteration $i$.

Once we have an estimate of the bounded mean of some truncated distribution, we use the following fact to translate that back into an estimate of the true underlying expected sufficient statistic.

**Proposition B.6** (Corollary 3.8 in [34]). *Let $\theta \in \Theta$. Suppose Assumption (S1) holds and that $q_\theta(S) > 0$. Then*

$$\|\mathbb{E}_{y \sim q_\theta^S}[T(y)] - \mathbb{E}_{x \sim q_\theta}[T(x)]\| \leq O\left(\log\left(\frac{1}{q_\theta(S)}\right)\right).$$

Proposition B.6 ensures that an good estimate of the truncated expected sufficient statistic is already a constant distance warm-start. However, in order to avoid $O(R)$-dependence on the prior radius, we iteratively refine our estimate. The following lemma analyzes the guarantees of one iteration of Algorithm B.2.

**Lemma B.7.** *Suppose Assumptions (S1) and (C3) holds. There is an $(\varepsilon, \delta)$-DP algorithm such that given truncated samples from $q_{\theta^\star}^{S_{warm}, R}$ and a prior parameter $\theta^{(0)}$ of distance at most $R \geq \Omega(\sqrt{m})$, estimates $\theta^\star$ up to distance $\alpha/\lambda$ for $\alpha \in (\Omega(\log(1/\rho)), R)$ with probability $1 - \beta$. Moreover, the algorithm has sample complexity*

$$O\left(\frac{R^2 \log(1/\beta)}{\alpha^2} + \frac{R\sqrt{m}\log(1/\delta)\log(1/\beta)}{\alpha\varepsilon}\right).$$

*Proof.* By a vector Bernstein inequality (Theorem A.9), taking

$$n \geq \Omega\left(\frac{R^2 \log(1/\beta)}{\alpha^2}\right)$$

samples ensure that the sample sufficient statistic is at most $\alpha/4$-distance from the expectation of the truncated sufficient statistic with probability $1 - \beta/2$. By standard Gaussian concentration inequalities (Theorem A.6), taking

$$n \geq \Omega\left(\frac{R\sqrt{m}\log(1/\delta)\log(1/\beta)}{\alpha\varepsilon}\right)$$

ensures the gaussian mechanism (Proposition A.2) adds noise of magnitude $O(\sigma\sqrt{m}\log(1/\beta)) \leq \alpha/4$ with probability $1 - \beta/2$. By Proposition B.6, this is then at most $O(\log(1/\rho)) = O(1)$-distance from the true expected sufficient statistic. Thus our estimate is distance at most $\alpha$ from the true expected sufficient statistic given the constant lower bound on $\alpha$ is sufficiently large. By the $\lambda$-strong convexity of the (untruncated) NLL function, the updated prior output by the moment matching oracle is $\alpha/\lambda$-distance from $\theta^\star$. $\qquad\square$

We are now ready to prove Lemma B.5.

*Proof of Lemma B.5.* Repeating the one-step algorithm from Lemma B.7 $\log(R/\sqrt{m})$ times ensures with iteratively halved accuracy parameters $\alpha = \lambda 2^{-i}R$ yields an estimate of the prior parameter of distance $O(\sqrt{m})$. This incurs a sample complexity blowup of $\tilde{O}(\log(R))$ for a total sample complexity of

$$\tilde{O}\left(\frac{\log(R/\beta)}{\lambda^2} + \frac{\sqrt{m}\log(1/\delta)\log(R/\beta)}{\lambda\varepsilon}\right).$$

We can then apply the one-step algorithm one last time but with $\alpha = O(\log(1/\rho))$. This incurs an additional sample complexity of

$$O\left(m\log(1/\beta) + \frac{m\log(1/\delta)\log(1/\beta)}{\varepsilon}\right).$$

We can re-use samples for each repetition. By simple composition (Theorem A.4), it suffices to incur another $\tilde{O}(\log(R))$ blow-up in sample complexity to preserve privacy. $\qquad\square$

## B.7 Coarse Bounding Box

As the final ingredient, we derive a private bounding box algorithm for $\theta^\star$ that translates to a $O(\sqrt{m}/\lambda)$-distance warm start. Combined with Appendix B.6, this completely removes the need for the simplifying Condition 3.3 from Sections 3.2 to 3.5.

The basis is a folklore result for Gaussians whose guarantees are stated by Karwa and Vadhan [30] but follows from prior works [10, 20, 47]. In particular, the idea is to learn an $\ell_\infty$ ball about $\mathbb{E}_{x \sim q_{\theta^\star}}[T(x)]$ by executing a private histogram algorithm on each of the coordinates. Translating the $\ell_\infty$ ball to a Euclidean ball about $\theta^\star$ yields a $O(\sqrt{m})$-distance warm-start. Lemma B.8 formalizes this idea. We in fact present a more general bounding box algorithm for truncated exponential families. For now, we can take the survival set to be all of $\mathbb{R}^d$ with survival mass $\rho = 1$.

**Lemma B.8.** *Suppose the exponential family has bounded covariances (Assumption (S1)), interiority (Assumption (S2)), and we have access to a moment-matching oracle (Assumption (C3)). Further suppose we are given truncated samples from $q_{\theta^\star}^S$ with survival mass $\rho$. There is an $(\varepsilon, \delta)$-DP algorithm that outputs some $\hat{\theta}$ and $\hat{\tau} = \mathbb{E}_{x \sim p_{\hat{\theta}}}[T(x)]$ such that $\|\hat{\theta} - \theta^\star\| \leq \tilde{O}\left(\frac{\sqrt{m}}{\eta\lambda} \log\left(\frac{1}{\rho\beta\delta\varepsilon}\right)\right)$ with probability $1 - \beta$. Moreover, the algorithm has sample complexity $n = \tilde{O}\left(\frac{m \log(1/\beta\delta)}{\varepsilon}\right)$ and running time $\mathrm{poly}(m, n, d)$.*

The proof of Lemma B.8 is deferred to Appendix B.7. As with all other algorithms we present, Lemma B.8 requires only truncated sample access for an exponential family.

We note that past works considered the multi-dimensional Gaussian case [40], but Lemma B.8 is the first to handle truncated exponential families.

**Pseudocode.** See Algorithm B.3 for pseudocode. As mentioned, it is a straightforward adaption of the Gaussian bounding interval algorithm stated by Karwa and Vadhan [30] to the multi-dimensional truncated exponential family case.

---

**Algorithm B.3:** Bounding Box

**Input:** truncated dataset $D = \{x^{(1)}, \ldots, x^{(n)}\}$, privacy parameters $\varepsilon, \delta \in (0, 1)$, bin length $s$
**Output:** estimator $\hat{\theta}$ for $\theta^\star$ and $\hat{\tau} = \mathbb{E}_{x \sim \hat{\theta}}[T(x)]$

1 **for** *coordinate* $i = 1, \ldots, m$ **do**
2      $[a, a + s] \leftarrow$ bin with largest estimated mass from the output of
     $\texttt{PrivateHistogram}(x_i^{(1)}, \ldots, x_i^{(n)}, s, \varepsilon/m, \delta/m)$        ▷ (Proposition B.9)
3      $\hat{\tau}_i \leftarrow a + s/2$

4 $\hat{\theta} \leftarrow \texttt{MomentMatch}(\hat{\tau})$

5 **return** $\hat{\theta}, \hat{\tau}$

---

**Analysis.** The bounding box algorithm crucially relies on a private histogram algorithm whose guarantees are stated by Karwa and Vadhan [30] but follows from the works of Bun, Nissim, and Stemmer [10], Dwork et al. [20], and Vadhan [47].

**Proposition B.9** (Histogram Learning; Lemma 2.3 in [30]). *Consider any countable distribution, say $p : \mathbb{Z} \to \mathbb{R}_+$, privacy parameters $\varepsilon, \delta \in (0, 1/n)$, error $\alpha > 0$, and confidence $\beta \in (0, 1)$. There is an $(\varepsilon, \delta)$-DP algorithm $\texttt{PrivateHistogram}$ that outputs estimates $\tilde{p}_i$ such that given*

$$n = \frac{8 \log(4/\beta\delta)}{\varepsilon\alpha} + \frac{\log(4/\beta)}{2\alpha^2},$$

*samples from $p$, then*

     *(i) $\|\tilde{p} - p\|_\infty \leq \alpha$ with probability at least $1 - \beta$ and*

     *(ii) $\Pr[\arg\max_k \tilde{p}_k = j] \leq np_j$.*

We are now equipped to derive our rough estimation algorithm.

**Theorem B.10.** *Let $X$ be a random vector such that the $i$-th centered coordinate $X_i - \mathbb{E}[X_i]$ is $(1, 1/\eta)$-subexponential for some $\eta \in (0, 1)$ and assume we have access to truncated samples from some survival set $S$ with mass $\rho > 0$. Discretize the real line using bins of length $s$ defined below.*

$$n := \frac{8 \log(4/\beta\delta)}{\varepsilon} + \frac{\log(4/\beta)}{2}, \qquad s := \frac{\log(2n/\rho\beta)}{2\eta}.$$

*Run the $(\varepsilon, \delta)$-histogram learner (Proposition B.9) on bins of length $s$ using the $i$-th coordinate of $n$ i.i.d. truncated samples and output the bin with the largest empirical mass along with its two adjacent bins. Then with probability at least $1 - \beta$, this interval of length $3s$ contains the untruncated mean $\mathbb{E}[X_i]$ of the $i$-th coordinate.*

*Proof.* Without loss of generality, consider the first coordinate $X_1$. By Proposition A.7,

$$\Pr_{x \sim X}\{|x_1 - \mathbb{E}[x_1]| \geq t\} \leq 2\exp\left(-\frac{t}{2\eta}\right) .$$

But for any event $\mathcal{E}$, we have $\Pr_{x \sim X^S}[\mathcal{E}] \leq \frac{1}{\rho}\Pr_{x \sim X}[\mathcal{E}]$. Hence

$$\Pr_{x \sim X^S}\{|x_1 - \mathbb{E}_{x \sim X}[x_1]| \geq t\} \leq \frac{2}{\rho}\exp\left(-\frac{t}{2\eta}\right) .$$

Thus with probability $1 - \beta$,

$$|x_1 - \mathbb{E}_{x \sim X}[x_1]| \leq \frac{\log(2/\rho\beta)}{2\eta} .$$

We claim that with probability $1 - \beta$, the central bin must contain a point within distance $s$ of $\mathbb{E}[X]$. Indeed, Let $J \subseteq \mathbb{Z}$ be the indices of bins which lie beyond distance $s$ of $\mathbb{E}[X]$. Then $\sum_{j \in J} p_j \leq \beta/n$ by the choice of $s$. Thus by Proposition B.9, the probability of outputting any such bin as the central bin is at most $\beta$.

Since the central bin must contain a point within distance $s$ of $\mathbb{E}[X]$, then by the definition of the bin length, the union of the central bin along with its adjacent bins must contain $\mathbb{E}[X]$, as desired. $\square$

We are now ready to prove Lemma B.8.

*Proof of Lemma B.8.* Privacy follows from the privacy of `PrivateHistogram` (Theorem B.10) and simple composition (Theorem A.4).

Consider a single coordinate from the sufficient statistic of a single sample $x \sim q_{\theta^\star}$, say $T(x)_1$ without loss of generality. Proposition A.8 assures us that $T(x)_1 - \mathbb{E}[T(x)_1]$ is $(1, 1/\eta)$-subexponential under the true measure $q_{\theta^\star}$. Thus the assumptions of Theorem B.10 hold and `PrivateHistogram` outputs an interval containing $\mathbb{E}[T(x)_1]$. Repeating this procedure with different coordinates from the same dataset yields an estimate $\hat{\tau}$ of $\mathbb{E}[T(x)]$ with $O(s)$ $\ell_\infty$-error or $O(s\sqrt{m})$ $\ell_2$-error. By the $\lambda$-strong convexity of the (untruncated) population NLL for $q_{\theta^\star}$, `MomentMatch` (Assumption (C3)) returns some $\hat{\theta}$ such that $\|\hat{\theta} - \theta^\star\| \leq O(s\sqrt{m}/\lambda)$ as desired. $\square$

## C  Omitted Details from Section 4

### C.1  Verifying Assumptions from Section 4.1

The isotropic Gaussian density function is given by

$$p(x; \mu) = \frac{1}{(2\pi)^{d/2}}\exp\left(-\frac{1}{2}\|x - \mu\|^2\right) = \frac{1}{(2\pi)^{d/2}}\exp\left(-\frac{1}{2}\|x\|^2\right)\exp\left(\mu^\top x - \frac{1}{2}\|\mu\|^2\right) .$$

Thus the parameter is given by the mean $\theta = \mu$ and the sufficient statistic is taken to be the identity $T(x) = x$. We take $\Theta = \mathbb{R}^d$.

**Statistical Assumptions.**  We first check that Assumption 2.2 holds.

- (S1) (Bounded Condition Number) For any $\mu \in \mathbb{R}^d$, the covariance of $\mathcal{N}(\mu, I)$ is the identity matrix. Hence $\lambda = 1$.

- (S2) (Interiority) There is a ball of radius 1 about every $\mu \in \mathbb{R}^d$.

- (S3) (Log-Concavity) Each $\mathcal{N}(\mu, I), \mu \in \mathbb{R}^d$ is a log-concave distribution.

- (S4) (Polynomial Sufficient Statistics) $T(x)$ is a polynomial of degree $k = 1$.

**Computational Subroutines.** Next we describe how to implement the subroutines specified in Assumption 2.3.

- (C1) (Projection Acess to Convex Parameter Space) We take $\Theta = \mathbb{R}^d$ so that the convex projection is the identity function.
- (C2) (Sample Access to Log-Concave Distribution) We can sample $z \sim \mathcal{N}(0, I)$ and $\mu + z$ is a sample from $\mathcal{N}(\mu, I)$.
- (C3) (Moment-Matching Oracle) If $\mathbb{E}[T(x)] = \tau$, then the corresponding parameter is simply $\mu = \tau$.

## C.2 Proof of Theorem 4.2

We now restate and prove Theorem 4.2.

**Theorem 4.2.** *Let $\varepsilon, \delta, \alpha, \beta \in (0, 1)$ and suppose that $\lambda I \preceq \Sigma^\star \preceq \Lambda I$. There is an $(\varepsilon, \delta)$-DP algorithm such that given samples from a Gaussian distribution $\mathcal{N}(0, \Sigma^\star)$, outputs an estimate $\widehat{\Sigma}$ satisfying $\|I - (\Sigma^\star)^{-\frac{1}{2}}\widehat{\Sigma}(\Sigma^\star)^{-\frac{1}{2}}\|_F \leq \alpha$ with probability $1 - \beta$. Moreover, the algorithm has sample complexity $n = \tilde{O}(\frac{d^{1.5} \log(\Lambda/\lambda\beta\delta)}{\varepsilon} + \frac{d^2 \log(1/\beta\delta)}{\varepsilon} + \frac{d^2 \log(1/\beta)}{\alpha^2} + \frac{d^2 \log(1/\beta\delta)}{\alpha\varepsilon})$ and running time* $\mathrm{poly}(n, d)$.

By scaling if necessary, we can work under the simplifying condition that $\lambda I \preceq \Sigma^\star \preceq \frac{1}{8}I$ for some $\lambda \in (0, 1/8)$. In Lemma C.2, we will see how to precondition the distribution so that $\lambda = \Omega(1)$.

We begin by verifying that Assumptions 2.2 and 2.3 hold in Appendix C.3, leading to the following corollary of Theorem 3.1.

**Lemma C.1.** *Let $\varepsilon, \delta, \alpha, \beta \in (0, 1)$ and suppose $\lambda I \preceq \Sigma^\star \preceq \frac{1}{8}I$. There is an $(\varepsilon, \delta)$-DP algorithm such that given samples from a Gaussian distribution $\mathcal{N}(0, \Sigma^\star)$, outputs an estimate $\widehat{M}$ satisfying $\|\widehat{M} - (\Sigma^\star)^{-1}\|_F \leq \alpha$ with probability $1 - \beta$. Moreover, the algorithm has sample complexity $n = \tilde{O}\left(\frac{d^2 \log(1/\beta\delta)}{\lambda^2\varepsilon} + \frac{d^2 \log(1/\beta)}{\lambda^4\alpha^2} + \frac{e^{O(1/\lambda^2)}d^2 \log(1/\beta\delta)}{\lambda^2\alpha\varepsilon}\right)$ and time complexity* $\mathrm{poly}(n, d)$.

As mentioned, we require a private preconditioning algorithm to avoid polynomial dependence on $1/\lambda$. We extend one such algorithm for Gaussians due to Biswas et al. [7] to the case of truncated Gaussians.

The idea is to truncate the data to a centered ball of radius $\frac{\sqrt{d}}{1-\rho}$ to preserve $\rho$ survival mass and then apply the following lemma.

**Lemma C.2.** *Let $\varepsilon, \delta, \alpha, \beta \in (0, 1)$ and assume $\lambda I \preceq \Sigma^\star \preceq \frac{1}{8}I$. Algorithm C.1 is an $(\varepsilon, \delta)$-DP algorithm such that given samples from a truncated Gaussian distribution $\mathcal{N}(0, \Sigma^\star, S)$ with survival probability $\rho > 0$, outputs an estimate $\widehat{\Sigma}$ satisfying $\Omega(\rho^2)I \preceq (\Sigma^\star)^{-\frac{1}{2}}\widehat{\Sigma}(\Sigma^\star)^{-\frac{1}{2}} \preceq O(\log(1/\rho))I$ with probability $1 - \beta$. Moreover, the algorithm has sample complexity $n = \tilde{O}\left(\frac{d^{1.5} \log(1/\lambda\rho\beta\delta)}{\varepsilon}\right)$ and runnning time* $\mathrm{poly}(n, d)$.

We present the proof of Lemma C.2 in Appendix C.4. By combining Lemmas C.1 and C.2, we obtain a proof of Theorem 4.2.

*Proof of Theorem 4.2.* By Lemma C.2, after truncating to $B_{\frac{\sqrt{m}}{1-\rho}}(0)$, Algorithm C.1 yields a constant error estimate of the true covariance matrix. By Lemma C.1, preconditioning further samples and executing the general algorithm for Gaussian covariances as an exponential family (Theorem 3.1) yields the desired result. $\qquad\square$

## C.3 Verifying Assumptions from Appendix C.2

As a reminder, we work under the simplifying condition that $\lambda I \preceq \Sigma^\star \preceq \frac{1}{8}I$. Let $|M|$ denote the determinant of a square matrix $M$. Let $M = \Sigma^{-1}$ denote the precision matrix of $\mathcal{N}(0, \Sigma)$. The zero-mean Gaussian density function parameterized by $M$ is given by

$$p(x; M) = \frac{1}{(2\pi)^{d/2}|M|^{-1/2}} \exp\left(-\frac{1}{2}x^\top M x\right).$$

The exponential family parameter is given by the precision matrix $\theta = M$ and the sufficient statistic is taken to be $T(x) = -\frac{1}{2}xx^\top$. We take the closed convex parameter space $\Theta = \{M \in \mathbb{S}_+^d : 7I \preceq M \preceq \frac{2}{\lambda}I\}$.

**Statistical Assumptions.** Once again, we must verify that the Assumption 2.2 is satisfied in order to apply Theorem 3.1. We require the following result by Lee, Mehrotra, and Zampetakis [35].

**Proposition C.3** (Lemma 9.1 in [35]). *Let $\lambda, \Lambda$ denote the smallest and largest eigenvalues of the covariance matrix $\Sigma = M^{-1} \succ 0$. Then*

$$\frac{\min(\lambda^2, \sqrt{\lambda})}{4} \cdot I \preceq \mathrm{Cov}_{x \sim p_M}[T(x), T(x)] \preceq 7\max(\lambda, \Lambda^2).$$

We are now ready to perform the verification.

(S1) (Bounded Condition Number) By Proposition C.3, for any $M \in \Theta$, the Fisher information is spectrally lower bounded by $\Omega(\lambda^2)$ and upper bounded by 1.

(S2) (Interiority) For $M^\star = (\Sigma^\star)^{-1} \in \Theta$, any $M' \in B_1(M)$ satisfies

$$7I \preceq M' \preceq (1/\lambda + 1)I \preceq \frac{2}{\lambda}I.$$

Hence $M' \in \Theta$ and we can take $\eta = 1$.

(S3) (Log-Concavity) Each $\mathcal{N}(0, M^{-1}), M \in \Theta$ is a log-concave distribution.

(S4) (Polynomial Sufficient Statistics) $T(x) = -\frac{1}{2}xx^\top$ is a polynomial of degree $k = 2$.

**Computational Subroutines.** We also describe how to implement the computational subroutines from Assumption 2.3.

(C1) (Projection Acess to Convex Parameter Space) For any symmetric matrix $M \in \mathbb{S}^d$, its projection onto $\Theta$ can be computed by solving a semi-definite program.

(C2) (Sample Access to Log-Concave Distribution) We can sample $z \sim \mathcal{N}(0, I)$ and $\Sigma^{\frac{1}{2}}\mu$ is a sample from $\mathcal{N}(0, \Sigma)$.

(C3) (Moment-Matching Oracle) If $\mathbb{E}[-\frac{1}{2}xx^\top] = \Sigma$, then the corresponding parameter is simply $\theta = -\frac{1}{2}\Sigma^{-1}$.

## C.4 Proof of Lemma C.2

Below, we restate and prove Lemma C.2.

**Lemma C.2.** *Let $\varepsilon, \delta, \alpha, \beta \in (0, 1)$ and assume $\lambda I \preceq \Sigma^\star \preceq \frac{1}{8}I$. Algorithm C.1 is an $(\varepsilon, \delta)$-DP algorithm such that given samples from a truncated Gaussian distribution $\mathcal{N}(0, \Sigma^\star, S)$ with survival probability $\rho > 0$, outputs an estimate $\widehat{\Sigma}$ satisfying $\Omega(\rho^2)I \preceq (\Sigma^\star)^{-\frac{1}{2}}\widehat{\Sigma}(\Sigma^\star)^{-\frac{1}{2}} \preceq O(\log(1/\rho))I$ with probability $1 - \beta$. Moreover, the algorithm has sample complexity $n = \tilde{O}\left(\frac{d^{1.5}\log(1/\lambda\rho\beta\delta)}{\varepsilon}\right)$ and runnning time $\mathrm{poly}(n, d)$.*

We emphasize that all algorithms we present require only truncated sample access for an exponential family. While there are private preconditioning algorithms without any dependence on $\Lambda/\lambda$, it is not clear if their guarantees hold for truncated samples.

**Pseudocode.** See Algorithm C.1 for pseudocode of the preconditioning algorithm. As mentioned, it is a straightforward adaptation of the Gaussian covariance estimation algorithm of Biswas et al. [7], with the main difference being the initial truncation step. Due to the bias introduced by truncation, this adapation is only able to achieve a constant-error approximation.

**Analysis.** We are given truncated sample access to a zero-mean Gaussian distribution and would like to privately learn its second moment up to constant relative spectral error.

Our proof requires the following facts about truncated statistics.

**Proposition C.4** (Lemma 5 in Daskalakis et al. [17]). *Let $\Sigma_S$ denote the covariance of the truncated Gaussian $\mathcal{N}(0, \Sigma, S)$ with survival mass $\rho > 0$. The following hold.*

(i) $\max_{i \in [n]} \|\Sigma^{-\frac{1}{2}} x^{(i)}\|$ *of $n$ truncated samples $x^{(i)}$ is $O\left(\sqrt{d}\log\left(\frac{nd}{\rho\beta}\right)\right)$ with probability $1 - \beta$.*

(ii) *The empirical covariance $\hat{\Sigma}_S$ of $n$ truncated samples satisfies $(1 - \alpha)\Sigma_S \preceq \hat{\Sigma}_S \preceq (1 + \alpha)\Sigma_S$ with probability $1 - \beta$ whenever $n \geq \tilde{\Omega}\left(\frac{d\log^2(1/\rho\beta)}{\alpha^2}\right)$.*

The following result allows us to relate the covariance matrix of a truncated Gaussian with its original covariance.

**Proposition C.5** (Lemma 6 in [17]). *Let $\Sigma_S^\star$ denote the covariance matrix of the truncated Gaussian distribution $\mathcal{N}(0, \Sigma^\star, S)$ with survival mass $\rho > 0$. Then*

$$\Omega(\rho^2)I \preceq (\Sigma^\star)^{-\frac{1}{2}} \Sigma_S^\star (\Sigma^\star)^{-\frac{1}{2}} \preceq O(\log(1/\rho))I.$$

We are now equipped to prove Lemma C.2.

*Proof of Lemma C.2.* Privacy follows by the guarantees of the Gaussian mechanism (Proposition A.2) and simple composition (Theorem A.4).

Following the presentation of Biswas et al. [7], we can assume without loss of generality we know that $I \preceq \Sigma^\star \preceq \kappa I$. This can be achieve by scaling the data by $1/\sqrt{\lambda}$ and taking $\kappa = 1/\lambda$. Then by Proposition C.5, $\Omega(\rho^2)I \preceq \Sigma_S^\star \preceq O(\kappa \log(1/\rho))I$. We begin with the preconditioner $A_0 := \frac{1}{\sqrt{\kappa}}I$. Assume inductively that we have $A_{i-1}$ such that

$$\Sigma_S^\star \preceq (A^{(i-1)})^{-1} \Sigma_S^\star (A^{(i-1)})^{-1} \preceq (2 - 2^{-i+1})\Sigma_S^\star + \kappa 2^{-i+1}I.$$

We would like to output some $A^{(i)}$ that satisfies the above induction hypothesis.

By Proposition C.4, the relative spectral error of the sample covariance is at most $1/8$ with probability $1 - \frac{\beta}{\Omega(\log(\kappa))}$ when

$$n \geq \tilde{\Omega}(d \log(\kappa/\rho\beta)).$$

Similarly, by Theorem A.10, the error due to the noise $Y$ added for privacy is at most $1/8$ with probability $1 - \frac{\beta}{\Omega(\log(\kappa))}$ when

$$\sigma \leq O\left(\frac{1}{\sqrt{d}\log(\kappa/\beta)}\right) \iff n \geq \Omega\left(\frac{d^{1.5}\log(\kappa/\rho\beta)\sqrt{\ln(1/\delta)}}{\varepsilon}\right).$$

The rest of the inductive step are simple calculations that can be found in Biswas et al. [7, Appendix B.3, arXiv version]. The only difference is that we replace the two spectral norm concentration bounds that Biswas et al. [7] used with the two listed above. After $v = O(\log(\kappa))$ iterations and conditioning on the concentration bounds, the inductive hypothesis implies that

$$\frac{1}{3}I \preceq A^{(v-1)} \Sigma_S^\star A^{(v-1)} \preceq I.$$

Finally, we pay a multiplicative spectral error of $\text{poly}(\rho)$ when translating the guarantees of the estimate of the truncated covariance to the true covariance.

$\square$

We remark that Ashtiani and Liaw [4, Theorem 5.4, arXiv version] give a polynomial-time $(\varepsilon, \delta)$-DP algorithm to estimate the covariance up to constant error and with sample complexity that is independent of the condition number. However, it is not clear that their algorithm works without change for *truncated* Gaussians, unlike the rest of our results (see Remark 3.2).

**Algorithm C.1:** Recursive Preconditioning

**Input:** $n$-sample truncated dataset $D$, privacy parameters $\varepsilon, \delta \in (0,1)$, initial spectral lowerbound bound $\lambda$, survival probability $\rho \in (0,1)$

**Output:** estimator $\hat{\Sigma}$ for covariance matrix $\Sigma^\star$

1   $\kappa' \leftarrow O\left(\frac{\log(1/\rho)}{\lambda \rho^2}\right)$

2   $\varepsilon' \leftarrow \frac{\varepsilon}{\log(\kappa')}$

3   $\delta' \leftarrow \frac{\delta}{\log(\kappa')}$

4   $\sigma \leftarrow O\left(\frac{d\sqrt{\log(1/\delta')}\log(nd/\rho\beta)}{n\varepsilon'}\right)$

5   $R \leftarrow \tilde{O}(\sqrt{d}\log(nd/\rho\beta))$

6   $A^{(0)} \leftarrow \frac{1}{\sqrt{\kappa'}}I$

7   **for** $i = 0, \ldots, v := O(\log(\kappa'))$ **do**

8      $w^{(j)} \leftarrow A^{(i)}x^{(j)}$ for $j \in [n]$           $\triangleright \|w^{(j)}\| \leq R$ w.h.p.

9      $w^{(j)} \leftarrow \text{proj}_{B_R(0)}(w^{(j)})$ for $j \in [n]$

10     $Z \leftarrow \frac{1}{n}\sum_j w^{(j)}(w^{(j)})^\top$         $\triangleright$ sensitivity $\Delta = \tilde{O}_{\rho,\beta}(d/n)$

11     $Y \leftarrow$ Gaussian matrix with symmetric entries $Y_{ij} \sim \mathcal{N}(0, \sigma^2)$

12     $Z^{(i+1)} \leftarrow S + Y$       $\triangleright$ Gaussian mechanism (Proposition A.2)

13     $U \leftarrow Z^{(i+1)} + \frac{1}{4}I$

14     $A^{(i+1)} \leftarrow U^{-\frac{1}{2}}A^{(i)}$

15   **return** $(A^{v-1})^{-1}Z^{(v)}(A^{(v-1)})^{-1}$

