# OpenReview forum: "Private Statistical Estimation via Truncation"
_NeurIPS.cc/2025/Conference — NeurIPS 2025 poster_

### Official Review · Reviewer_LDZ7 · 2025-06-27

**Clarity:** 2
**Significance:** 2
**Originality:** 3
**Rating:** 3
**Confidence:** 3

**Summary:**

This paper proposed a differentially private statistical estimation framework by using techniques from truncated statistics. They then apply their framework to exponential family distributions. A distribution $q_\theta$ (on $R^d$) from the exponential families takes the form $q_\theta(x) \propto h(x) \exp(\theta^\top T(x))$ where $\theta \in R^m$ is the unknown parameter, $h : R^d \to R_+$ is the base measure, $T : R^d \to R^m$ is a sufficient statistic.
This paper obtains an algorithm with sample complexity that depends linearly on $m$ (dimension of sufficient statistics), polynomially on $1/\alpha\varepsilon$, $1/\varepsilon$, $1/\alpha^2$, exponentially on the condition number. Their algorithm runs in time poly(n, m, d) (n is the number of samples and d is the dimension of each sample). They instantiate their algorithms for Gaussian mean estimation (with known covariance matrices) and Gaussian covariance estimation. For both problems, their algorithm gets the right sample complexity dependence on the dimension.
Their algorithmic framework first chooses a "survival set" and truncates data to that set, and then minimizes the empirical truncated NLL function using DP-SGD.

**Questions:**

- How does the sample complexity in Theorem 3.1 compare with the non-private one (i.e. what can be achieved without privacy)?
- The framework shares some similarity with the clip-and-noise method. Can you elaborate on their differences?
- Can the 1-d algorithm from [MSU20] be extended to high-dimensional settings?

**Ethical Concerns:**

["NO or VERY MINOR ethics concerns only"]

**Final Justification:**

I think the connection between DP statistics and truncated statistics is nice and natural, but the exponential dependence on the condition number as well as its application to gaussian estimation makes the significance of the contribution questionable. The authors claim one can use private preconditioner to reduce the condition number, but this is usually highly non-trivial and problem specific. Given this, I decide to keep my rating.

**Limitations:**

yes

**Quality:**

2

**Strengths And Weaknesses:**

**Strengths:**
- This paper made a nice observation that results/techniques from truncated statistics could be made use of for DP estimation. I think this connection is very natural and might have great potential.
- This paper is generally well written.

**Weakness:**
- The results for Gaussian estimation are actually suboptimal and the claim of (near-)optimality for Gaussian estimation in the paper is overstated. For Gaussian mean estimation (Theorem 4.1), e.g., let's consider the regime when beta is a sufficiently small constant. Then the optimal sample complexity is $d/\alpha^2 + d/\alpha\varepsilon + \log(1/\delta)/\varepsilon$, while the sample complexity in Theorem 4.1 is $d/\alpha^2 + d\log(1/\delta)/\alpha\varepsilon$. For Gaussian covariance estimation (Theorem 4.2), their sample complexity depends on the condition number. Although the dependence is only logarithmic, the optimal sample complexity should not depend on the condition number at all.
- It's not very clear how good the sample complexity stated in Theorem 3.1 is, without enough context about what can be achieved non-privately. I think there should be more detailed discussion. E.g. where does the exponential dependence on the condition number arise?

---

> ### Author Rebuttal · Authors · 2025-07-30
>
> We thank the reviewer for recognizing the important conceptual connection between truncation and differential privacy. While this seems intuitive, we are the first to formalize this connection and design algorithms with rigorous utility guarantees and sample complexity bounds. We are hopeful that this can open the door to more DP algorithmic developments in future works, obtained via truncation.
>
> Our main result is a private learner for exponential families that only accesses the sensitive data after a truncation pre-processing step. As a bonus, our algorithms go through even if the provided data has already undergone truncation!
> When specialized to Gaussian mean estimation, our techniques indeed yields the correct sample complexity up to logarithmic factors in the lower-order terms. We will certainly clarify this in the camera-ready version.
> For Gaussian covariance estimation, if we are ok with dropping the bonus mentioned above, we can use any private pre-conditioning algorithm to avoid all dependence on the condition number. This logarithmic dependence comes up only under the restricted sample access since it is not clear if previous private Gaussian preconditioning algorithms can extend to the truncated setting.
>
> Please find below the responses to specific questions:
> 1. Using (non-private) SGD to minimize the log-likelihood of exponential families in our setting yields a sample complexity of $O(\frac{m}{\alpha^2} \cdot \text{poly}(\kappa))$ where $m$ is the dimension of the parameter space, $\alpha$ is the $\ell_2$-error parameter, and $\kappa$ is the condition number (see e.g. Theorem 4.3 in [42] from references section).
>
>     We note that the main difference is the dependence on $\kappa$, which is necessary as it arises from using techniques from the truncated statistics literature to lower bound the strong convexity parameter of the *truncated* log-likelihood function (see e.g. [34] in references section). Indeed, this exponential dependence appears even for (non-privately) estimating truncated exponential families.
> 2. One difference between truncation and clipping is that minimizing the truncated log-likelihood function yields a consistent estimator as long as the survival set has positive mass, e.g., 50%. This truncation of half of the data introduces biases that we then remove using our algorithms. In contrast, the clipping radius must depend on the concentration of the underlying distribution, as we must ensure no point is clipped with high probability so that no truncation bias is introduced. We commit to adding a discussion about this in the final version.
> 3. If we understand correctly, the framework from [36] in the references section is not generally suitable for estimating multivariate statistics, as it relies on a simple-hypothesis-testing formulation that does not fully capture private estimation for multivariate statistics. Specifically, (private) hypothesis testing and parameter estimation depend on the dimension in different ways.
>
> We commit to clarifying the discussions above in the camera-ready version.

---

> > ### Comment · Reviewer_LDZ7 · 2025-08-06
> >
> > Thanks for the detailed answers. For my first question, if I understand it correctly, non-privately, the sample complexity depends polynomially on the condition number, while the sample complexity of your algorithm (theorem 3.1) has exponential dependence on the condition number. This limitation seems to put negative influence on the significance of the general technique.

---

> > > ### Author Response · Authors · 2025-08-07
> > >
> > > We thank the reviewer for clarifying their concern.
> > >
> > > While the sample complexity of (non-privately) learning exponential families without truncation using SGD depends polynomially on the condition number,
> > > the sample complexity of (non-privately) learning **truncated** exponential families using SGD depends exponentially on the condition number.
> > > Our general SGD learner can privately learn truncated exponential distributions, and hence also has exponential dependence on the condition number.
> > > We would like to remark that for many exponential families (e.g., Gaussians), it is possible to pre-condition the samples so that the condition number is a constant.
> > > Then, our general SGD learner can be applied to the preprocessed samples to avoid this exponential dependence.

---

### Official Review · Reviewer_dfJz · 2025-06-30

**Clarity:** 3
**Significance:** 4
**Originality:** 4
**Rating:** 5
**Confidence:** 3

**Summary:**

The paper considers DP statistical estimation, i.e. estimating the parameter of a distribution given samples from that distribution with differential privacy. Normally, differential privacy requires a bound on the sensitivity of some function (e.g. the sample mean) of the samples, which is at odds with sampling from unbounded distributions. The authors consider methods from the non-private statistics literature for working with samples restricted to a "truncation set". They give an algorithm which obtains the optimal sample complexity dependence on the dimension $m$, privacy parameter $\epsilon$, and error $\alpha$ of $O(m(\frac{1}{\epsilon} + \frac{1}{\alpha^2} + \frac{1}{\alpha \epsilon})$ in polynomial time.

Their algorithm has a number of components and associated proofs. First, they give a procedure for padding/truncating a dataset so all samples appear in the truncation set and the dataset has fixed size, and with high probability the final dataset matches n samples from the distribution restricted to the truncation set, which allows them to work with the replace adjacency. Next, they assume a warm-start which they show can be obtained by a recursive mean estimation algorithm. Given the warm-start they run DP-SGD using the negative log likelihood ratio of the samples as a loss. They show this is strongly convex under an assumption about the condition number of the covariance of the sufficient statistics, hence DP-SGD has a known bounded error for this loss. They also show how to obtain an unbiased estimator of gradients of this loss with bounded gradient norms.

**Questions:**

* Theorem 3.4 is somewhat outdated. For example, the authors could maybe use results from Section 5 of https://arxiv.org/pdf/2005.04763, which removes an unnecessary $\log^2 (n/\delta)$ factor from BST14, and also only requires a single pass which would alleviate the problem discussed in Section 3.3 "Uniform Convergence of Empirical Likelihood". Have the authors considered this / is it straightforward to integrate these to simplify the analysis and slightly improve the bounds?

**Ethical Concerns:**

["NO or VERY MINOR ethics concerns only"]

**Final Justification:**

I remain happy with the paper, and the authors also nicely addressed my only stated weakness, which was minor to begin with.

**Limitations:**

Yes

**Quality:**

4

**Strengths And Weaknesses:**

Strengths:
* First paper to get the right dependence on the privacy parameter $\epsilon$ and error parameter $\alpha$ for general exponential families without requiring boundedness or dimension 1.
* The paper retrieves the best rates for Gaussian mean and covariance estimation using a more generally applicable algorithm.
* The technical aspects of the paper are reasonably novel; e.g. optimization of the negative LLR is pretty different from the private selection-based approaches used for the optimal results on Gaussians.
* Presentation in the paper is good; the key steps are frequently outlined throughout the paper, the assumptions are clearly stated, the technical barriers and how they are dealt with are summarized nicely.

Weaknesses:
* Some of the results from the DP optimization literature which are used as a black box are outdated and the paper might be improved "for free" by using more recent results; see Questions for more detail.

---

> ### Author Rebuttal · Authors · 2025-07-30
>
> We thank the reviewer for recognizing the novelty and technical contributions of our work, as well as sharing the reference to more recent work on DP stochastic convex optimization.
>
> The ``iterative localization’’ technique from [FKT20] does indeed directly optimize the population likelihood in one pass, under the additional assumptions that the likelihood is smooth and the condition number is not too large. For exponential families where we have a private preconditioning algorithm (e.g., Gaussians), this is indeed applicable to our setting and can simplify the proof as well as yield a logarithmic improvement in the bounds.
> However, in general, for badly conditioned exponential families, this additional requirement on the condition number bound is more nuanced.
> We will be sure to include this discussion in the camera-ready version.
>
> ---
>
> [FKT20] Feldman, V., Koren, T., & Talwar, K. (2020). Private stochastic convex optimization: Optimal rates in linear time

---

> > ### Comment · Reviewer_dfJz · 2025-08-04
> >
> > Thanks for your response, I see now this does not allow an immediate improvement. I remain happy with the paper and will maintain my positive score.

---

### Official Review · Reviewer_REUx · 2025-07-02

**Clarity:** 4
**Significance:** 2
**Originality:** 2
**Rating:** 4
**Confidence:** 3

**Summary:**

The paper presents a new framework for differentially private (DP) statistical estimation using truncated statistics, focusing on unbounded exponential family distributions. It proposes an efficient DP algorithm for estimating parameters, like Gaussian mean and covariance, with optimal sample complexity. The method truncates data to control sensitivity, corrects bias using DP-SGD and maximum likelihood estimation, and adapts existing algorithms for truncated exponential families.

**Questions:**

- At line 173, the article supposes that the distributions are absolutely continuous. Is this with respect to Lebesgue measure? If so, what is the technical step that is blocking with more general measures? Conceptually, shouldn’t it work as long as the empirical log-likelihood is almost surely differentiable (in order for first-order methods to make sense) and that the other assumptions are satisfied?
- I did not see (but I may have missed it) an assumption on the boundedness of the set in which we are searching the parameter. However, without such an assumption, how do you guarantee that the loss is Lipschitz, which is necessary for the privacy of DP-SGD? Indeed, as the losses are supposed strongly convex, the gradients are coercive, and ensuring their boundedness typically requires a priori information on the location of the optimal parameter.
- Linked to the last question, can the authors detail how the sample complexity depends on the size of the a-priori set in which we expect to find the optimal parameter?
- In terms of pure technical contributions, is the main technical advancement Lemma 2.1?

**Ethical Concerns:**

["NO or VERY MINOR ethics concerns only"]

**Final Justification:**

After reading the author’s response, I’ve decided to retain my positive rating of the article, as I believe it’s a valuable contribution.

**Limitations:**

Yes

**Quality:**

3

**Strengths And Weaknesses:**

Strengths
- The paper’s is well written and clear, making it easy to follow.
- The framework for combining truncation with DP is innovative, and recovers optimal sample complexities for Gaussian estimation.
- The generality of the article make it applicable to many real-world scenarios

Weaknesses
- It can be hard to tell what’s new versus what’s built on existing results.
- Moving basic exponential family facts to the appendix and diving deeper into technical contributions would help clarify the paper’s novelty.
- Also, the paper doesn’t present a single numerical experiment, which weakens its practical validation and leaves questions about real-world performance.

---

> ### Author Rebuttal · Authors · 2025-07-30
>
> We thank the reviewer for their careful reading, suggestions on the presentation, and for appreciating our results. We will incorporate these suggestions into the camera-ready version of our paper.
>
> Our main conceptual contribution is the connection between truncation and differential privacy. While this seems intuitive, we are the first to formalize this connection and design algorithms with rigorous utility guarantees and sample complexity. We believe this can open the door to more practical algorithmic developments in future work.
>
> We emphasize that our techniques lead to the first algorithm for privately estimating unbounded high-dimensional truncated families, whereas prior works focus on bounded or 1-parameter families.
>
> Please find the responses to specific questions below:
> 1. We thank the reviewer for pointing out the additional generality of our technique. We indeed assumed absolute continuity with respect to the Lebesgue measure for simplicity to ensure the log-likelihood is differentiable. Our algorithm goes through as long as the latter holds. We will be sure to mention this in the camera-ready version.
> 1. We demonstrate how to privately obtain a rough estimate of the parameter (warm start) in Appendix B without any prior bounds. Previous works demonstrated how to achieve this for Gaussian distributions, and we extended the techniques to truncated exponential families. Once we have a rough estimate of the parameter, our algorithm chooses the survival set to be a bounded set of appropriate radius (depending on the warm start), which automatically yields a bound on the gradient norm of the *truncated* log-likelihood function. This is a key benefit of the truncation framework and optimizing the truncated log-likelihood.
> 1. Our general algorithm for exponential families does not have any dependence on prior bounds of the parameter (please see previous response).
> 1. Technically, Lemma 2.1 shows that it suffices to design DP algorithms for truncated distributions by designing a ``stable’’ reduction from (untruncated) distributions to truncated ones. We hope that this reduction can lead to other DP algorithms obtained via truncation. As noted in Section 1.1, another technical contribution that may be of independent interest beyond our work is the improved uniform convergence result for exponential families (Lemma 3.6), which improves upon prior works by a quadratic factor in the error parameter.

---

> > ### Comment · Reviewer_REUx · 2025-08-04
> >
> > Thank you for your detailed and thoughtful rebuttal. I appreciate the clarifications provided, especially regarding the technical assumptions and the scope of your theoretical contributions. Your responses have addressed my main concerns, and I find the overall contribution of the paper to be solid and valuable. I will maintain my positive score for this submission.

---

### Official Review · Reviewer_RRzL · 2025-07-06

**Clarity:** 3
**Significance:** 4
**Originality:** 3
**Rating:** 4
**Confidence:** 2

**Summary:**

This paper introduces a novel framework for differentially private (DP) statistical estimation via data truncation, addressing the fundamental challenge of sensitivity control when the data support is unbounded. The authors propose a truncation-enhanced DP-SGD algorithm that enables efficient parameter estimation for exponential-family distributions while achieving near-optimal sample complexity. The theoretical contributions are significant.

**Questions:**

1. In practice, how should the survival set, i.e. the ball radius, be chosen?
2. Could you elaborate on warm-start and preconditioning?
3. What failure probability beta is reasonable for real-world data sets?

**Ethical Concerns:**

["NO or VERY MINOR ethics concerns only"]

**Limitations:**

The same as above.

**Paper Formatting Concerns:**

No.

**Quality:**

3

**Strengths And Weaknesses:**

Strengths
1) Novel and interesting approach. Using truncated data to bound sensitivity in DP estimation is both theoretically sound and practically valuable.
2) Technical innovations. Compared with [3], the complexity of estimating Gaussian statistics is reduced.
3) Theoretical proofs. The authors make a substantial effort to establish rigorous theoretical results.

Weaknesses
1) Although the paper provides strong theoretical results for Gaussian data, it is unclear how to apply the method to real-world datasets. For example, how to choose the survival set in practice is not well explained; how to warm-start and preconditioning; and what failure probability beta is reasonable.
2) The algorithm is more complex than classical DP methods, so it would be helpful to include visualizations to better illustrate the practical benefits over classical DP. It remains unclear whether the increased computational complexity is acceptable for large-scale datasets.
3) Many key results rely on a set of strong and potentially difficult-to-verify assumptions. It remains unclear whether these assumptions are realistically satisfiable in practical scenarios. If these assumptions are violated, how the performance or guarantees of the proposed algorithm would degrade?
4) Although many theoretical results are provided, no numerical experiments or empirical validation are included to demonstrate practical performance, convergence behavior, and related aspects.

---

> ### Author Rebuttal · Authors · 2025-07-30
>
> We thank the reviewer for their thoughtful and constructive feedback and for appreciating our results.
>
> Our work is primarily a theoretical study, and our main goal is to formalize and rigorously analyze the connection between truncation and differential privacy. While truncation should intuitively synergize with privacy, our work is first to establish a systematic framework with precise utility guarantees and sample complexity bounds. We view this conceptual link as the main message of our paper, and we believe it can open the door to more practical algorithmic developments in future work.
>
> To the best of our knowledge, our techniques lead to the first algorithm for privately estimating unbounded high-dimensional truncated families, whereas prior works focus on bounded or 1-parameter families.
>
> We emphasize that we adopt standard exponential family regularity assumptions and the assumptions customary in truncated statistics analyses. We state the assumptions explicitly and quantify how constants depend on them.
>
> Please find below the responses to specific questions:
> 1. In practice, we would ideally choose the survival set to contain a large constant fraction of the distribution, e.g., 90%. This can be a hyperparameter which is tuned. We would like to point out that one benefit of the truncated statistics framework is that the estimator is consistent regardless of the survival mass. This is a feature that is not shared by standard clipping operations to guarantee privacy.
> 2. We provide a generic warm-start algorithm in the appendix. Intuitively, it suffices to privately estimate the parameter of the *truncated* distribution, which will be a biased estimate of the true parameter but suffices as a warm start. This is an easier task since the truncated distribution is a bounded distribution.
>
>     We also provide a pre-conditioning algorithm in the appendix. Again, the idea is to privately estimate the parameters of the *truncated* distribution.
> 3. In practice, a reasonable rule of thumb is to choose the failure probability beta to be a small constant, e.g., 5%. It may be helpful to think of beta as the significance level of a statistical study which should be set according to the field of study.

---

### Note · Authors · 2025-08-14

We thank all reviewers for their thorough evaluation and appreciation of our work. We commit to including the appropriate discussions in the final version.

Our central contribution is introducing the first efficient framework for differentially private (DP) estimation of unbounded distributions by formally connecting it with truncated statistics. As the reviewers noticed, this yields the first DP algorithm for estimating unbounded high-dimensional exponential families that is nearly minimax-optimal, up to logarithmic factors in the lower-order terms, when specialized to the cases of Gaussian mean and covariance estimation.

Regarding the sample complexity dependence of the general learner on the condition number, we wish to clarify that this is an artifact inherited from the non-private truncated statistics literature itself, and not a limitation introduced by our techniques. Crucially, as we detail for Gaussians, this dependence can be eliminated via private pre-conditioning, making our framework more broadly applicable.

Our work provides a foundational and principled blueprint for algorithm design where data has unbounded support, a persistent challenge in DP. We believe this conceptual link, along with our improved uniform convergence guarantees, can lead to further improvements in DP algorithm design.

---

### Decision · Program_Chairs · 2025-09-17

**Decision:**

Accept (poster)

**Comment:**

This work proposes algorithms for differentially private (DP) statistical estimation by leveraging recent algorithmic techniques from the area of truncated statistics. The main application is new DP algorithms for certain exponential families. The reviewers appreciated the contribution. While the idea of using some kind of truncation is fairly standard in the context of DP estimation, the connection to the truncated statistics literature is a useful observation. Some concerns were raised on certain extraneous dependencies (e.g., on condition number) in the sample complexity of the resulting algorithms, even for the Gaussian case. While the concrete new implications themselves may not be as strong as desired, the conceptual contribution of the work is interesting and I am inclined to recommend borderline acceptance.